# Insight into the RssB-Mediated Recognition and Delivery of σ^s^ to the AAA+ Protease, ClpXP

**DOI:** 10.3390/biom10040615

**Published:** 2020-04-16

**Authors:** Dimce Micevski, Kornelius Zeth, Terrence D. Mulhern, Verena J. Schuenemann, Jessica E. Zammit, Kaye N. Truscott, David A. Dougan

**Affiliations:** 1Department of Biochemistry and Genetics, La Trobe Institute for Molecular Science, La Trobe University, Melbourne 3086, Victoria, Australia; jmicevski85@hotmail.com (D.M.); 16111322@students.latrobe.edu.au (J.E.Z.); 2Department of Protein Evolution, Max-Planck-Institute for Developmental Biology, D-72076 Tübingen, Germany; kzeth@ruc.dk (K.Z.); verena.schuenemann@iem.uzh.ch (V.J.S.); 3Department of Science and Environment, Roskilde University, DK-4000 Roskilde, Denmark; 4Department of Biochemistry and Molecular Biology, The University of Melbourne, Parkville 3010, Victoria, Australia; tmulhern@unimelb.edu.au

**Keywords:** RssB, SigmaS, AAA+ protease, ClpX, X-ray structure, adaptor protein

## Abstract

In *Escherichia coli*, SigmaS (σ^S^) is the master regulator of the general stress response. The cellular levels of σ^S^ are controlled by transcription, translation and protein stability. The turnover of σ^S^, by the AAA+ protease (ClpXP), is tightly regulated by a dedicated adaptor protein, termed RssB (Regulator of Sigma S protein B)—which is an atypical member of the response regulator (RR) family. Currently however, the molecular mechanism of σ^S^ recognition and delivery by RssB is only poorly understood. Here we describe the crystal structures of both RssB domains (RssB_N_ and RssB_C_) and the SAXS analysis of full-length RssB (both free and in complex with σ^S^). Together with our biochemical analysis we propose a model for the recognition and delivery of σ^S^ by this essential adaptor protein. Similar to most bacterial RRs, the N-terminal domain of RssB (RssB_N_) comprises a typical mixed (βα)_5_-fold. Although phosphorylation of RssB_N_ (at Asp58) is essential for high affinity binding of σ^S^, much of the direct binding to σ^S^ occurs via the C-terminal effector domain of RssB (RssB_C_). In contrast to most RRs the effector domain of RssB forms a β-sandwich fold composed of two sheets surrounded by α-helical protrusions and as such, shares structural homology with serine/threonine phosphatases that exhibit a PPM/PP2C fold. Our biochemical data demonstrate that this domain plays a key role in both substrate interaction and docking to the zinc binding domain (ZBD) of ClpX. We propose that RssB docking to the ZBD of ClpX overlaps with the docking site of another regulator of RssB, the anti-adaptor IraD. Hence, we speculate that docking to ClpX may trigger release of its substrate through activation of a “closed” state (as seen in the RssB-IraD complex), thereby coupling adaptor docking (to ClpX) with substrate release. This competitive docking to RssB would prevent futile interaction of ClpX with the IraD-RssB complex (which lacks a substrate). Finally, substrate recognition by RssB appears to be regulated by a key residue (Arg117) within the α5 helix of the N-terminal domain. Importantly, this residue is not directly involved in σ^S^ interaction, as σ^S^ binding to the R117A mutant can be restored by phosphorylation. Likewise, R117A retains the ability to interact with and activate ClpX for degradation of σ^S^, both in the presence and absence of acetyl phosphate. Therefore, we propose that this region of RssB (the α5 helix) plays a critical role in driving interaction with σ^S^ at a distal site.

## 1. Introduction

Bacteria are constantly exposed to a range of changing environmental conditions. Their ability to adapt to each of these changes is central to their survival. Although some responses (e.g., the heat-shock response) are highly tailored to specific environmental stresses, others (e.g., the general stress response) offer broad cross-protection to the cell [1,2,3]. In *Escherichia coli* the general stress response is best characterized by the transition from exponential growth to stationary growth and as such is regulated by a specialized sigma factor, known as the starvation sigma factor, σ^s^ (also called σ^38^ or RpoS) which is responsible for the induction of up to 100 gene products. Importantly, this response not only protects cells from nutrient starvation but also from hyperosmotic stress, acid and alkaline stress, heat and cold shock. Given σ^s^ plays such an important role in the cell, it’s not surprising that its activity and levels are tightly controlled, not only at the transcriptional and translational levels, but also at the post-translational level via ATP-dependent proteolysis [2,4,5]. 

In *E. coli*, cytosolic protein degradation is performed by five different AAA+ (ATPase associated with a variety of cellular activities) proteases (ClpXP, ClpAP, HslUV, Lon, and FtsH). Typically, these proteolytic machines recognize their substrates via a degradation sequence (degron), which is generally located at the termini of the protein [6,7,8,9]. For the most part, each degron (and hence protein substrate) is recognized by a single AAA+ protease, although on occasions a single degron may be recognized by multiple AAA+ proteases [10]. In some cases, an additional component (termed an adaptor protein) is recruited by the AAA+ protease for recognition of a specific substrate (or set of substrates). Some adaptor proteins are essential for the recognition and delivery of their substrates (e.g., ClpS and the turnover of N-degron substrates by ClpAP), while others simply alter the rate of substrate turnover (e.g., SspB and the turnover of SsrA-tagged proteins by ClpXP) [11,12,13,14,15]. In the case of σ^s^, a specialized adaptor protein RssB, also referred to as SprE [16,17], is not only required for direct recognition of the substrate but also required for its preparation and delivery to the AAA+ protease, ClpXP [18,19]. In addition to the adaptor protein RssB, the levels of σ^s^ are also controlled by an additional group of proteins, termed anti-adaptors (e.g., IraD, inhibitor of RssB activity after DNA damage), all of which inhibit the activity of RssB [20,21]. Each anti-adaptor is induced in response to a specific stress, and currently three anti-adaptor proteins have been identified. Although the precise mechanism of action of these anti-adaptors remains unclear, all are proposed to inhibit σ^s^ turnover via direct interaction with RssB [22,23,24]. 

RssB is a response regulator (RR), and as such is composed of an N-terminal receiver (REC) domain and a C-terminal output domain (also known as an effector domain). In contrast to most RRs (which contain a DNA binding domain) RssB is linked to a protein binding domain, and as such represents one of the small groups of RRs (~1% of all RRs). Nevertheless, like most RRs, RssB is phosphorylated on a conserved Asp residue (Asp58) within the N-terminal REC domain. However, in contrast to most RRs the physiological significance of RssB phosphorylation, and its effect on the output domain remains unclear [18,25]. Despite this, the in vitro phosphorylation of RssB by acetyl phosphate (AcP) appears to enhance both the recognition of σ^s^ (by RssB) and its turnover (by ClpXP). Currently however, despite extensive research on this topic from several groups, little is known about the mechanism of substrate recognition by RssB, its engagement with ClpX or its mechanism of substrate delivery to ClpX(P). For example, which region(s) of RssB are responsible for substrate interaction? How does phosphorylation of the N-terminal domain of RssB fine-tune substrate interaction? What are the molecular determinants of RssB docking to ClpX? Is ClpX activated by this docking and how does substrate handover occur? Previously we proposed (by virtue of a conserved C-terminal motif termed the ClpX binding region (XBR) found in both RssB and SspB) that RssB delivered its substrate via docking of the adaptor to the N-terminal zinc binding domain (ZBD) of ClpX [12]. 

Here we have performed a wide range of structural (X-ray crystallography and small angle X-ray scattering (SAXS)) and biochemical (pull-down and degradation assays, immunoprecipitation, peptide library screen, and chemical crosslinking) experiments to develop a model for the RssB-mediated delivery of σ^s^ to the ClpXP. Our data demonstrate that the effector domain of RssB is necessary for interaction with both σ^s^ and ClpX. However, this domain alone is insufficient for substrate delivery to ClpX, demonstrating that the N-terminal domain of RssB also plays a significant role in this process. Consistently, our in vitro data show that phosphorylation of RssB_N_ (at Asp58) modulates σ^s^ binding and turnover. In addition, based on analysis of the RssB mutant (R117A), we have identified Arg117 as a key residue that regulates σ^s^ interaction. Importantly, this residue is not directly involved in substrate recognition, as σ^s^ binding to R117A is almost completely restored in the presence of AcP. Therefore, we propose that Arg117 is a key “switch” residue, required to stabilize a substrate “engagement” conformation of RssB, which is likely related to the phosphorylation-dependent conformation of RssB. Significantly, Arg117 is located on the mobile α5 helix of RssB_N_, which is directly linked to the C-terminal domain that makes direct contact to σ^s^. Remarkably, despite significantly reduced ability to recognize σ^s^, the R117A mutant was able to facilitate σ^s^ turnover by ClpXP. Based on these data, we speculate that docking (of RssB) to the N-terminal ZBD of ClpX, activates ClpX for σ^s^ recognition. From our peptide library analysis, we propose that both the ZBD of ClpX and the anti-adaptor IraD dock to a common or partially overlapping site within RssB and speculate that the interaction of the ZBD with RssB may represent an elegant method to facilitate the timely release of σ^s^ into the hexameric pore of ClpX. Hence, we propose that RssB docking to σ^s^ is not only essential for activation of the substrate, but also required for docking to the ZBD of ClpX, thereby co-ordinating release of σ^s^ and the activation of ClpX for substrate recognition and translocation into ClpP. 

## 2. Materials and Methods 

### 2.1. Cloning

Constructs encoding RssB mutants (D58E, D58K, K108A, K108R, K108D, RE/AA, and R117A) were generated by site-directed mutagenesis [26] using either *pHUE/RssB* or *pET32a/RssB* as template DNA. Refer to supplementary material for primer sequences (Appendix A) and the final plasmid constructs generated (Appendix A). All clones were verified by nucleotide sequencing. 

### 2.2. Protein Purification and Size Exclusion Chromatography

His_6_-tagged ClpX and ClpP were overexpressed in *E. coli* and purified as described previously [12]. Untagged σ^S^, RssB (residues 1–337), RssB_C_ (residues 131–337) and RssB_N_ (residues 1–128) (wild type and specific point mutants) were generated using the Ub-fusion system [27] and purified essentially as described previously [24], using a combination of IMAC (immobilized metal ion affinity chromatography) and preparative grade size exclusion chromatography (SEC) using a HiLoad 16/600 Superdex 200 to separate monomeric and dimeric RssB. All columns were pre-equilibrated in chilled GF buffer (20 mM Tris-HCl pH 7.5, 10 mM MgCl_2_, 0.1 mM EDTA, 1 mM DTT, 140 mM NaCl, 5% (*v*/*v*) glycerol, 0.005% (*v*/*v*) Triton X-100).

### 2.3. In Vitro Degradation Assays

The in vitro σ^S^ degradation assays were performed essentially as described [19] with minor modifications. Briefly, ClpX (1 μM), ClpP (1 μM), monomeric RssB (80 nM) and 20 mM of the phospho donor, acetyl phosphate (AcP) were pre-incubated in degradation buffer (20 mM Tris-HCl, 140 mM NaCl, 10 mM MgCl_2_, 0.1 mM EDTA, 5% (*v*/*v*) glycerol, 0.005% (*v*/*v*) Triton X-100, 1 mM DTT, pH 7.5) at 30 °C, with 1 μM σ^S^ [28]. All reactions were initiated with the addition of 2 mM ATP and samples collected at the specified time-points. Samples were separated using SDS-PAGE and visualized using Coomassie Brilliant Blue (CBB) staining. Quantitation of protein bands was performed using either GelAnalyzer or GelEval.

### 2.4. Glutaraldehyde Cross-Linking 

Studies probing transient interactions between σ^s^, the ClpX ZBD, RssB and the various RssB derivatives were carried out using 35 μg of each respective protein in the presence of 0.005% (*v*/*v*) glutaraldehyde (GA) and 1x cross-linking buffer (20 mM Hepes-KOH pH 7.5, 140 mM NaCl, 10 mM MgCl_2_, 0.1 mM EDTA, 5% (*v*/*v*) glycerol, 0.005% (*v/v*) Triton X-100, 1 mM DTT). Proteins were pre-incubated in the absence of GA for 2 min. Upon initiating GA cross-linking, 50 μL samples were collected at specified time points and diluted into 0.3 M Tris-HCl pH 8.8. Samples were prepared in Laemmli buffer (1×) and separated by SDS-PAGE. Cross-linking products were visualized by immunodecoration with the appropriate antisera.

### 2.5. Peptide Library

To identify RssB peptides interacting with the *E. coli* ClpX ZBD (*ec*ClpX_ZBD_), a commercially derived RssB peptide library (JPT Peptide Technologies) composed of 13-mer peptides (overlapping by 10 residues) immobilized to a cellulose membrane (see Appendix A for peptide sequences), was panned with *ec*ClpX_ZBD_, essentially as described previously [29] with some modifications. Pure *ec*ClpX_ZBD_ (2.5 μM) was incubated with the RssB peptide library in MP2 buffer (15.7 mM Tris-HCl pH 7.6, 100 mM KCl, 20 mM MgCl_2_, 5% (*w*/*v*) sucrose, 0.05% (*v*/*v*) Tween20) for 30 min. with gentle shaking at room temperature (RT). Upon completion of the previous incubation, the peptide library was washed with ice cold TBS (50 mM Tris-HCl pH 8.0, 137 mM NaCl, 2.7 mM KCl) for 30 s, and then positioned on 4 pieces of pre-soaked (cathode buffer: 25 mM Tris-HCl pH 9.2, 40 mM 6-aminohexanoic acid, 0.01% (*w*/*v*) SDS) 3MM Whatman blotting paper. The sandwich was then layered with a methanol (100%) and Anode 1 (A1) buffer (30 mM Tris and 20% (*v*/*v*) methanol) soaked PVDF membrane, 4 × 3 MM Whatman blotting paper soaked in A1 buffer and 4 × 3 MM Whatman blotting paper soaked in Anode 2 (A2) buffer (300 mM Tris, 20% (*v*/*v*) methanol). Bound protein was then transferred (6×) to a PVDF membrane at a current of 1 mA × area (cm^2^) of the peptide library for the following times 30, 30, 20, 20, 30, and 30 min. The PVDF membrane was replaced after each transfer, while the 3MM Whatman blotting sheets were only replaced after every second transfer. Bound proteins were detected by immunodecoration using anti-ClpX antisera (1:1000 dilution in 3% (*w/v*) skim milk powder/TBS + 0.05% (*v*/*v*) Tween-20 (TBS-T)), washed in 1 × TBS-T before being incubated with the appropriate secondary antibody conjugated with HRP, and visualized using either a ChemiGenius2 (Syngene, Cambridge, U.K.) or GelDoc™ XR + (Bio-Rad, Hercules, U.S.A.) imagining system and images captured using Genesnap (Syngene) or QuantityOne (Bio-Rad) respectively. Figures were generated by overlaying 6 membrane images with adjusted transparency (100%, 50%, 33%, 25%, 20%) to ensure that all images had equal contribution to the overall result.

### 2.6. In Vitro ‘Pull-Down’ Experiments

To examine the interaction of RssB (full-length or individual domains) with purified ZBD or σ^s^, in vitro “pull-down” experiments were performed essentially as previously described [24,30]. Briefly, Ni-NTA agarose beads (QIAGEN, Hilden, Germany) (bed volume (BV), 25 μL) were equilibrated with 10 BV of equilibration buffer (50 mM Tris-HCl pH 8.0, 300 mM NaCl, 10 mM imidazole) followed by 5 BV of Wash Buffer C (20 mM Hepes-KOH pH 7.5, 100 mM KOAc, 10 mM MgOAc, 10% (*v*/*v*) glycerol, 10 mM imidazole, 0.5% (*v*/*v*) Triton X-100). Proteins (25 μg) were pre-incubated in ice cold Wash Buffer C (30 min. on a rotating wheel), then applied to Ni-NTA agarose beads and incubated as previously described [24]. Samples were separated by centrifugation (800 g, 1 min, 4 °C), the beads were resuspended in 5 BV of Wash Buffer C and transferred to individual MoBiTec (Molecular Biologische Technologie) columns. The Ni-NTA agarose beads were then washed with 10 BV of Wash Buffer D (20 mM Hepes-KOH pH 7.5, 100 mM KOAc, 10 mM MgOAc, 10% (*v*/*v*) glycerol, 20 mM imidazole, 0.25% (*v*/*v*) Triton X-100) and residual buffer removed via centrifugation (800 g, 1 min, 4 °C). Bound proteins were eluted via centrifugation (800 g, 1 min, 4 °C) using 1 BV of Elution Buffer (50 mM Tris-HCl pH 8.0, 300 mM NaCl, 250 mM imidazole). Samples were separated via SDS-PAGE and proteins visualized by staining with Coomassie Brilliant Blue or transferred to a PVDF membranes for visualization by immunodecoration with the appropriate anti-sera. Quantitation of protein bands was performed using GelAnalyzer or GelEval.

### 2.7. Co-Immunoprecipitation (Co-IP)

For co-IP experiments, specific IgGs were conjugated to Protein A Sepharose (PAS) Fast-Flow beads (GE Healthcare), the beads were prepared as described [31]. Anti-serum (diluted 2-fold with 0.2 M KPi pH 7.5) was added to pre-equilibrated (in 0.1 M KPi pH 7.5) PAS beads and incubated (on a rotating wheel) for 60 min at 4 °C. Unbound antibodies were removed from the bead suspension by centrifugation (800 g, 3 min, 4 °C) before the beads were rinsed with 10 BV of 0.1 M Na-borate pH 9.0. The beads were then incubated (30 min, 4 °C with gentle rotation on spinning wheel) with 10 BV of fresh 0.1 M Na-borate pH 9.0 containing 14 mM Dimethyl-Pimilimidate dihydrochloride (DMP, Sigma), then the cross-linking process was repeated with the addition of fresh 14 mM DMP. Conjugated IgG-PAS beads were then recovered via centrifugation (800 g, 5 min), rinsed with 10 BV of 1 M Tris-HCl pH 7.5 and incubated in 10 BV of 1 M Tris-HCl pH 7.5 for 2 h at 4 °C (gentle rotation on spinning wheel). Finally, conjugated IgG-PAS beads were rinsed with 10 BV of 10 mM Tris-HCl pH 7.5 and stored on ice pending co-IP experiments.

Co-IP experiments were performed using purified recombinant proteins (0.14 nmol) and 20 μL (settled BV) of cross-linked PAS beads (see above) essentially as described [32]. In brief, proteins were pre-incubated in 1x co-IP buffer (20 mM Tris-HCl pH 7.5, 10 mM MgCl_2_, 140 mM KCl, 0.1 mM EDTA, 5% (*v/v*) glycerol, 0.005% (*v/v*) Triton X-100, 1 mM DTT) and 10 mM AcP (Sigma-Aldrich) at 4 °C for 30 min (gentle rotation on spinning wheel). Pre-incubated protein samples were then added to cross-linked PAS beads and incubated for 60 min at 4 °C (gentle rotation on spinning wheel). Unbound proteins were removed via centrifugation (800 g, 2 min, 4 °C), beads washed 5x with 5 BV of 1x co-IP buffer and bound proteins recovered by centrifugation (16,060 g for 5 min) with 1 BV of 50 mM glycine pH 2.5. Eluted fractions were neutralized in 250 mM Tris-HCl pH 8.0 and analyzed by SDS-PAGE.

### 2.8. Crystallization of RssB_N_ and RssB_C_ and Diffraction Data Collection

Initial sitting-drop crystallization trials of both domains were performed with the Honeybee 961 robot (Genomic solutions) using 13 commercial high throughput screens on 3:1 Corning 3550, 96-well plates. Drops mixed from 0.4 μL of protein solution and 0.4 μL of reservoir solution were equilibrated against 75 μL of precipitant solution at 20 °C. Crystals of different shape appeared under a variety of conditions that were further reproduced under hanging-drop conditions. Crystals of RssB_N_ suitable for the X-ray data collection were obtained with a solution containing 20% (*w/v*) PEG 20,000 and 0.1 M HEPES buffer, pH 7.5. Crystals of RssB_C_ suitable for X-ray data collection were obtained with a mixture containing 10% (*w/v*) PEG 8000, 0.2 M Li_2_SO_4_ and 0.05 M MES buffer, pH 6.5.

### 2.9. Structure Determination and Refinement

The structure of RssB_N_ was solved by molecular replacement using the coordinates of 2PL1 as starting model. The structure of RssB_C_ was solved by multiple anomalous diffraction methods using protein crystals soaked with Pt-salts. Protein crystals were flash-frozen in the presence of 20% (*v/v*) glycerol. All x-ray crystallographic data were collected at beamline PX10 at the Swiss Light Source (SLS, Villigen, Switzerland). The crystals were kept at 100 K during data collection, and a 360° dataset was collected. Data were processed with the XDS/XSCALE [33] program package, and the structure was solved using the SHARP/autoSHARP program package [34,35]. Initial model building was performed in Arp/Warp and Buccaneer [36,37]. Subsequent improvement of models was achieved using the PHENIX program package in combination with Coot for the construction of the models [38,39]. The structure was analyzed using PROCHECK [40], and the figures were prepared using the PyMOL program package.

### 2.10. Small Angle X-ray Scattering (SAXS) of RssB, σ^S^ and the RssB/σ^S^ Complex

SAXS experiments were performed on the SAXS/WAXS beamline at the Australian Synchrotron. Preliminary experiments, performed with Trx-RssB as a static sample, indicated the presence of high MW aggregates, which interfered with the analysis. To overcome this problem, SEC in-line with the synchrotron SAXS was used to fractionate samples and obtain scattering from the isolated species, rather than mixtures. These SAXS data were collected using an established in-line gel filtration chromatography protocol [41]. Sample injections of Trx-RssB (20 nmol), σ^S^ (35 nmol) and complexed RssB- σ^S^ (at a 1:1 molar ratio, 12.5–15 nmol) in the presence of AcP (10 mM) were loaded onto a Superdex 200 16/600 preparative column (GE Healthcare), pre-equilibrated in RssB gel-filtration buffer (20 mM Tris-HCl pH 7.5, 10 mM MgCl_2_, 0.1 mM EDTA, 1 mM DTT, 140 mM NaCl, 5% (*v*/*v*) glycerol, 0.005% (*v*/*v*) Triton X-100), at a flow rate of 0.5 mL/min. For each run, 600 detector images were collected as 2 s exposures every 2.1 s. The images were analyzed as averages of blocks of five sequential exposures using the SAXS15ID software (Australian Synchrotron). Each image series was converted to 120 individual *I(q)* vs. *q* profiles, where *I(q)* is scattered X-ray intensity as a function of the momentum transfer vector *q* = (4πsinθ)/λ, where the scattering angle is 2θ and the X-ray wavelength is λ (1.0332 Å). The *q* range over which intensities were collected was 0.016-0.524 Å^−1^. SAXS data were analyzed using the ATSAS suite of programs (version 2.4) [42]. Masses of the individual proteins and protein complexes were estimated by Porod analysis using AUTOPOROD, while the radius of gyration (*R_g_*) was estimated by Guiner analysis using AUTORG. Pair distance vector distribution functions *P(r)* were calculated using AUTOGNOM, which also yielded the maximum dimension (*D_max_*) of the scattering particle and the forward scattering intensity, *I(0)*. Final data sets were generated by averaging data corresponding to the respective peaks where *R_g_*, *D_max_* and the MW were consistent. Using the *P(r)* curves as input, 10 DAMMIF models were generated for each species, which were then averaged and filtered using DAMAVER [43]. Rigid body refinement of thioredoxin (Trx), RssB_N_, RssB_C_, and σ^S^ with inclusion of linkers was attempted using BUNCH [44]. Nonetheless, due largely to the length of the interdomain linkers, inconsistent domain arrangement solutions emerged. Therefore, both the N- and C-domains of RssB were manually fitted into the shape envelope to provide a qualitative model consistent with Trx at the N-terminus of the RssB_N_ to determine the possible position of σ^S^ in the Trx-RssB- σ^S^ complex.

## 3. Results

### 3.1. The RssB/σ^s^ Complex Docks to the ZBD of ClpX

Initially, to validate the idea that RssB and SspB use a common docking platform on ClpX for delivery of its substrates to ClpXP, we performed a series of adaptor competition experiments in which we monitored the RssB-mediated turnover of σ^s^, in the absence or presence of SspB (Figure 1). Consistent with our initial hypothesis, the RssB-mediated turnover of σ^s^ was inhibited by the addition of SspB (Figure 1a, filled symbols). Indeed, in the presence of 15 µM SspB (the highest concentration that we tested, Figure 1a, filled circles) the half-life (t½) of σ^s^ was increased by ~9-fold (i.e., from ~2.5 min in the absence of SspB, to ~23 min). To ensure that the SspB-mediated inhibition of σ^s^ turnover was not simply due to steric hinderance of the pore of ClpX (by SspB) we also monitored the RssB-mediated turnover of σ^s^ in the presence of the XBR peptide from SspB (SspB_XBR_) (Figure 1b). Consistent with docking of RssB/σ^s^ to the XBR docking site on the ZBD of ClpX, increasing concentrations of the SspB_XBR_ inhibited the turnover of σ^s^ (Figure 1b). Collectively these data suggest that interaction with the adaptor docking platform on ClpX (i.e., the ZBD) is essential for the RssB-mediated delivery of σ^s^. However, during the early stages of this study the structure of RssB from *Pseudomonas aeruginosa* was reported (PDB: 3EQ2) and it became apparent that the XBR motif of *^Pa^*RssB was integral to the protein fold and hence unlikely to be available for interaction with ClpX. Therefore, we modified our hypothesis regarding the mechanism of σ^s^ delivery to ClpX, although based on the above results, we maintained a focus on the role the ZBD in this process. Initially, we monitored the ClpP-dependent turnover of σ^s^ in the presence of either wild type ClpX (Figure 1c, top panel) or a ClpX mutant (_∆ZBD_ClpX) which lacked the ZBD (Figure 1c, lower panel). As expected, σ^s^ was rapidly degraded (t½ ~2.5 min) by wild type ClpXP in the presence of RssB (Figure 1c, top panel), while deletion of the ZBD of ClpX (_∆ZBD_ClpX) completely inhibited the RssB-mediated turnover of σ^s^ (Figure 1c, lower panel). Importantly, deletion of this domain does not affect the turnover of adaptor-independent substrates such as SsrA-tagged proteins [12,13] demonstrating that _∆ZBD_ClpX retains full unfoldase and ClpP-docking activities. Collectively, these data demonstrate that the ZBD of ClpX is essential for the turnover of σ^s^, likely as a docking platform for RssB. Of note, following consumption of σ^s^, full-length ClpX was cleaved into a smaller fragment (Figure 1c, lanes 5 and 6, upper panel). Interestingly, and consistent with the findings of Houry and colleagues [45], this “clipping” appears to involve the N-terminal region of ClpX as no such “clipping” was observed for _∆ZBD_ClpX (Figure 1c, lanes 5 and 6, lower panel). Next, we asked the question, can the ZBD alone inhibit the RssB-mediated degradation of σ^s^, and if so, which component does it interact with? To address these questions, we performed a series of competition assays, in which the RssB-mediated turnover of σ^s^ by wild type ClpXP was monitored in the presence of increasing concentrations of the ZBD of ClpX (Figure 1d). Consistent with Figure 1a, and docking of RssB to the ZBD of ClpX, the turnover of σ^s^ was inhibited upon addition of increasing concentrations of ZBD (Figure 1d, filled symbols). These data clearly demonstrate that the ZBD of ClpX is sufficient for interaction with RssB (and/or σ^s^), however, it still remained unclear which component(s) was directly involved in the interaction, or indeed how these protein(s) were recognized by the ZBD.

### 3.2. The C-Terminal Domain of RssB (RssB_C_) Docks Directly to the ZBD

Given the interaction between ClpX and RssB (or σ^s^) is either transient and/or low affinity [19], we decided to examine a possible interaction between the ZBD and RssB using chemical cross-linking. Specifically, we incubated the ZBD with or without RssB, in the presence of GA. Initially as a control, we examined the crosslinking of the His_6_-tagged ZBD alone (Figure 2a, lanes 1–6). As expected [15], rapid dimerization of the ZBD was observed in the presence of GA (Figure 2a, lanes 1–6). These data validate the experimental set-up of our system. Next, we monitored the formation of a complex between His_6_-tagged ZBD (using anti-His antisera) and RssB (Figure 2a, lanes 7–12). Consistent with a specific interaction between ClpX and RssB a crosslinked product was observed when the ZBD was incubated together with RssB (Figure 2a, lanes 7–12). Significantly, this crosslink was absent from control experiments, i.e., ZBD alone (Figure 2a, lanes 1–6) and more importantly the MW of the crosslinked band (~45–50 kDa) was equivalent to complex containing a single copy of RssB (~38 kDa) together with a dimer of the ZBD (~12 kDa). To verify the presence of RssB within this crosslink, the same reaction was probed with anti-RssB antisera (Figure 2b, lanes 7–12). Taken together, these data suggest that (in the absence of σ^s^) RssB can still interact with the ZBD of ClpX.

Next, in order to further dissect the interaction between the ZBD and RssB, we again performed chemical crosslinking of the ZBD, however in this case we monitored crosslinking in the presence of either RssB_C_ or RssB_N_. Consistent with Figure 2a, rapid dimerization of the ZBD was observed in the absence of RssB (Figure 2d). Interestingly, dimerization of the ZBD appears to be enhanced in the presence of RssB (either full length or C-domain) (Figure 2a,d, respectively, lanes 10–12). More importantly, addition of RssB_C_ also resulted in the appearance of two additional crosslinked bands (Figure 2c, ~30 and 40 kDa), which appear to contain both RssB_C_ (Figure 2c) and the ZBD (Figure 2d). Based on the apparent MW of the crosslinked complexes, we speculated that the smaller complex (~ 30 kDa) represents a heterodimeric complex of RssB_C_ crosslinked to a single ZBD (within the RssB_C_- ZBD_2_), while the higher MW complex (~40 kDa) represents RssB_C_ crosslinked to the ZBD dimer (RssB_C_-ZBD_2_). Significantly, no cross-linking was observed for ZBD, in the presence of RssB_N_ (Figure 2c, lanes 3–8). Finally, in order to identify a potential ZBD docking site on RssB we examined the binding of the ZBD to 13-mers from RssB using cellulose bound peptide library. Analysis of these binding data identified a total of 8 “strong” binding peptides, found in three regions of RssB (Figure 2e), one was located within the N-terminal domain (peptides 20–23) and two within the C-terminal domain (peptides 81–82 and 90–91), which may represent docking sites for ZBD. Intriguingly, the putative RssB_XBR_ motif (peptide 109) was not recognized by the ZBD. Consistent with this finding, the RssB_XBR_ peptide (in contrast to the SspB_XBR_ peptide, see Figure 1b) was unable to inhibit the ClpXP-mediated turnover of σ^s^ (data not shown). Notably, alignment of the “strong” binding peptides revealed a putative consensus binding motif for the ZBD (LKxh, where h = small hydrophobic and x = any amino acid, see Figure 2f), which is highly similar to the XBR motif of SspB [12,46]. Therefore, based on these data, we propose that RssB docks to the ZBD of ClpX (in a competitive manner to SspB), not through the putative C-terminal XBR motif of RssB, but rather via an alternative XBR-like site, possibly defined by peptides 81–82 or 90–91 (discussed later).

### 3.3. Structure of the C-Terminal Phosphatase Domain of RssB

To gain structural insight into how RssB might interact with its various different co-factors (ClpX, anti-adaptors or substrate) we attempted to solve the structure of RssB (full-length and individual domains) either alone or in complex with several different partner proteins. Although we were unsuccessful in crystallizing full-length RssB in complex with a partner protein, we did solve the structures of both RssB domains. The structure of the RssB_C_ was solved at 2.1 Å by multiple isomorphous replacement (see Appendix A). The RssB_C_ domain structure is a mixed α/β-fold with five α-helices and 11 β-strands and can be further divided into a mediator/connector domain (residues 131–163) connecting RssB_N_ with RssB_C_ and the actual C-terminal protein phosphatase (PP2C) domain (residues 164–337). The N-terminal part or connector domain of RssB_C_ comprises three small α-helices (α6–α8) which are arranged with approximate angles of 90 degrees to each other (see Figure 3a). These helices are connected to the core domain by salt bridges, hydrogen bonds and hydrophobic interactions, but the conformation of the very first α6-helix is likely misaligned due to the absence of the structuring RssB_N_ domain. The α8-helix connects the mediating α-helical elements with the C-proximal and structurally conserved phosphatase domain, which consists of 11 β-strands and two α-helices. The core structure of this domain is formed by a β–sandwich comprising two closely interacting β-sheets. One-half of this sandwich domain is formed by the anti-parallel β-sheet of five β-strands in the order β8-β9-β10-β11-β14 (Figure 3a). The two long helices (α9 and α10) connect the β9 and β10 strands and attach to this side of the beta sandwich domain. The second anti-parallel β-sheet of the sandwich domain is formed by six β-strands in the following sequence: β6-β7-β16-β15-β12-β13. The two β-sheets are stabilized through a hydrophobic core structure formed by conserved aromatic and aliphatic residues.

Next we searched for structural homologs of this domain using the DALI server (http://ekhidna2.biocenter.helsinki.fi/dali/). From this analysis we found the *E. coli* RssB-IraD complex to be the most similar structure, with an r.m.s.d. of 2 Å (for 205 aligned residues, PDB-entry: 6OD1). The second closest structure with an r.m.s.d. of 3 Å is the RssB protein from *P. aeruginosa* which was crystallized in different crystal forms either as the full-length protein or an individual domain (PDB-entries: 3F79, 3F7A, 3EQ2 and 3ES2, respectively). In superposition, this structure exhibits a surprisingly weak sequence identity of 16% and shows two additional helices connecting the last two β-strands (see Figure 3c). The conserved residues (between *^Ec^*RssB and *^Pa^*RssB) are mostly located in the hydrophobic interface but also include two positively charged surface exposed patches, one of which (Figure 3b, right panel) is centered around the putative phosphatase site that corresponds to the peptides 90 and 91 from the peptide library and hence may play a role in docking to the ZBD of ClpX.

RssB_C_ also shares structural similarity with RsbX, another protein phosphatase from *Bacillus subtilis*. Although *^Bs^*RsbX shared an r.m.s.d. of 2.8 Å with *^Ec^*RssB_C_ (for 155 aligned residues; PDB-entry: 3W41), it only shares 9% sequence identity with *^Ec^*RssB_C_ in structural superposition [47]. Interestingly, the active site of *^Bs^*RsbX contains a metal ion which is coordinated by three Asp residues, however only one of these Asp residues (Asp204) is conserved in *^Ec^*RssB. The RssB_C_ also shows structural homology to a variety of other phosphatase domains in the PDB with r.m.s.d. of ~3 Å and low structural sequence identity of ~10%. The superposition of RssB_C_ onto the serine/threonine phosphatase from *Streptococcus agalactiae* as representative of this class of phosphatases is shown in Figure 3d.

### 3.4. The N-Terminal Domain of RssB Adopts the Fold of a Two Component System Regulator

The structure of RssB_N_ was solved to 2.05 Å by molecular replacement using the PhoP response regulator structure (PDB-entry: 2PL1) as a search model. Overall, the structure consists of five repetitive (αβ)-elements which form a parallel β-sheet (β2–β1–β3–β4–β5–) sandwiched by two helices (α1 and α5) on one side and three helices (α2–α4) on the opposite side (Figure 4a). A structural search of the PDB identified unpublished structures of *E. coli* RssB_N_ (PDB-entry: 3EOD) and *P. aeruginosa* RssB (PDB-entry: 3F7A) to be the most similar models with an r.m.s.d. of 0.4 Å and 1.3 Å, respectively. The next most similar target was the *E. coli* RssB-IraD complex (PDB-entry: 6OD1). Superposition of RssB_N_ onto the RssB/IraD structure revealed an r.m.s.d. of 1.4 Å. While the core structure of this domain is well preserved (with the exception of small deviations around Asp58), there were clear changes in the conformation of the α5 helix across the structures. Interestingly, in contrast to most RRs this helix is somewhat unique as it is predicted to be the structural motif that connects the N- and C-terminal domains of RssB via an extended coiled coil structure (PDB-entry: 3F7A). From analysis of the B-factors of RssB_N_ (which resemble molecular flexibility) it became obvious that the last helix (α5), in particular, shows higher mobility (see Appendix A). This is supported by the observation of limited hydrophobic contacts between this helix and the core domain structure. The reason for these high B-factors may either be functional or may have occurred due to the crystal packing where the two monomers are packed in a non-biological orientation. Importantly, mutations in this region have a dramatic effect on the regulation of substrate binding (discussed below).

Similar to several REC domains, our structure of RssB_N_ formed a dimer in the asymmetric unit. However, in contrast to many RRs the RssB_N_ dimer showed an anti-parallel orientation, which is mainly stabilized by β-strand augmentation via the terminal β2-strands. This orientation is different to the biologically active dimer of several RRs including PhoP from *E. coli.* It is also distinct from the parallel arrangement observed in the structure of full-length *P. aeruginosa* RssB protein, where the opposite face (formed by α4-β5-α5, here termed the 4-5-5 interface) contact one another. Therefore, to examine if the conserved 4-5-5 interface of *E. coli* RssB was involved in dimerization, we modelled the dimer of *^Ec^*RssB_N_ using the dimeric, full-length *^Pa^*RssB structure (Figure 4c). Using this template, the dimeric RssB_N_ model only yielded a small interacting surface area. This small interface may explain why the two domains interact only weakly in solution and may form alternative dimerization sites. Nevertheless, consistent with a role for the 4-5-5 interface in *^Ec^*RssB dimerization, mutation of L106 was previously shown to stabilize dimer formation as determined by chemical cross-linking experiments [24]. To examine the possibility that a region equivalent to the coiled-coil region of *^Pa^*RssB also contributes to the weak dimerization of *^Ec^*RssB, we compared the primary sequence of both proteins. Consistent with secondary structure predictions and the recent structure of RssB (in complex with IraD [23]), *^Ec^*RssB lacks most of the key coiled-coil residues found in *^Pa^*RssB (Appendix A). Collectively, these data suggest that weak dimerization of *^Ec^*RssB likely occurs through the 4-5-5 interface but does not involve formation of a coiled-coil domain.

Next we examined which residues on the surface of RssB_N_ were conserved. From this analysis we identified two conserved regions (see Figure 4b). As expected, the first conserved region was formed by the loops L1, L3, and L5, which are located at the distal end of the molecule. This conserved patch includes Asp58 (the site of phosphorylation) and is surrounded by additional charged residues (see Figure 4b and c; Glu14, Asp15, and Glu16) which together coordinate a magnesium ion. The second conserved region forms a groove (Figure 4b), which centers on the C-terminal part of the protein (including the β5 strand). This groove includes Lys108, which forms salt bridges with Asp58 and Glu14 and is involved in transmitting the phosphorylation signal from the REC domain to the C-terminal output domain (see later).

### 3.5. The C-Terminal Domain of RssB is Required for σ^s^ Docking

Next we asked the question, how does RssB interact with σ^s^? To address this question, we examined the interaction of σ^s^ with RssB (full length and individual domains) using a range of in vitro techniques. Initially we monitored the interaction between RssB and σ^s^ by co-immunoprecipitation (co-IP) using anti-σ^s^ antisera (Figure 5a). As expected, immunoprecipitation of wild type RssB was observed in the presence of σ^s^ (Figure 5a, lane 2). Importantly, the level of RssB recovered (in the presence of σ^s^), was significantly higher than in the absence of σ^s^. The small recovery of RssB (in the absence of σ^s^) is likely due to a weak non-specific interaction of RssB with the beads to which the antisera was immobilized (Figure 5a, lane 1). Next we examined the ability of RssB_N_ (Figure 5a, lanes 3 and 4) or RssB_C_ (Figure 5a, lanes 5 and 6) to interact with σ^s^ and determined the relative importance of each domain by quantitating four independent co-IP experiments (Figure 5b). Similar to wild type RssB, and consistent with an important role for the C-terminal domain in σ^s^ interaction, RssB_C_ was co-immunoprecipitated in the presence of σ^s^. Importantly, the level of RssB_C_ recovered (in the presence of σ^s^) was significantly greater than in the absence of σ^s^ (Figure 5b). In contrast to RssB_C_, little-to-no RssB_N_ was recovered by co-IP using anti-σ^s^ antisera, suggesting that the N-terminal domain plays only a minor direct role in the interaction. Overall, these data clearly show that RssB_C_ plays a significant direct role in σ^s^ binding, however the apparent binding affinity of RssB_C_ (with σ^s^) is substantially compromised given much less RssB_C_ was recovered by co-IP (in comparison to wild type RssB). Hence, other regions (or conformations) of RssB are likely required for direct interaction with σ^s^. Consistent with these data, phosphorylation of RssB (on the N-terminal domain) enhances σ^s^ binding (see Figure 6).

To validate these findings, we performed a series of experiments in which the bait was reversed. In this case, we immobilized Trx-H_6_-RssB (wild type, RssB_N_ or RssB_C_) to Ni-NTA-beads and incubated each column with untagged σ^s^ (Figure 5c). As expected, σ^s^ was recovered from the pulldown using full length RssB (Figure 5c, lane 3). Importantly, reversing the bait reduced the level of non-specific binding of RssB. Consistent with the co-IP experiments, a significant amount of σ^s^ was also recovered from column bearing immobilized RssB_C_ (Figure 5c, lane 7), while only a very small amount of σ^s^ was recovered from the column bearing immobilized RssB_N_ (Figure 5c, lane 5). Taken together, these data validate a direct role for the C-terminal effector domain of RssB in σ^s^ recognition (Figure 2b). Finally, to confirm the identification of RssB_C_ as the primary docking site for σ^s^, we performed a series of non-specific cross-linking experiments using GA, in which σ^s^ was incubated in the absence or presence of RssB_C_. In order to identify σ^s^ specific crosslinks, we monitored the crosslinked proteins using anti-σ^s^ antisera (Figure 5d). Consistent with the co-IP and affinity pull-down experiments, a specific cross-link product (~50 kDa), accumulated in the presence of σ^s^ and RssB_C_ (Figure 5d, lanes 8–12), which was absent from the σ^s^ alone experiment (Figure 5d, lanes 1–6). Importantly, the MW of this crosslinked product (~50 kDa) was equivalent to the theoretical MW of a heterodimeric complex of σ^s^ and RssB_C_.

### 3.6. Mutations in RssB_N_ Regulate Phosphorylation Dependent σ^s^ Binding

Based on the findings above, the effector domain (RssB_C_) is primarily responsible for direct interaction with σ^s^, however this domain alone, only appears to account for about one third of the total binding activity/affinity of σ^s^ (in comparison to wild type RssB). Furthermore, the receiver domain alone, also contributes little to the direct binding of σ^s^, yet phosphorylation of this domain appears to play a significant role in the interaction. This suggests that substrate binding to RssB_C_ is either significantly enhanced (or stabilized) by phosphorylation of the N-terminal receiver domain or alternatively that recognition of σ^s^ extends beyond docking to RssB_C_ and likely includes a region of RssB, which is only exposed in the full-length protein and can be enhanced by phosphorylation of the N-terminal domain (such as the linker region between the two domains). However, a systematic in vitro analysis of the effect of phosphorylation and events that contribute to phosphorylation dependent conformational changes in RssB have yet to be examined in detail, as such the importance of RssB phosphorylation remains controversial. Therefore, we compared the rate of σ^s^ turnover in vitro in the absence or presence of AcP (Appendix A). Consistent with published findings, the addition of AcP enhanced the RssB-mediated turnover of σ^s^. The apparent K_m_ for the turnover of σ^s^ by unphosphorylated RssB was calculated to be ~150 nM, which decreased ~3-fold (K_m_ ~58 nM) in the presence of phosphorylated RssB (RssB~P). These data clearly demonstrate, that although phosphorylation of RssB is not essential for the turnover of σ^s^ it does alter its affinity for the substrate. Next, we generated a series of mutations within RssB_N_ to dissect the mechanism of activation by phosphorylation. As expected, mutation of Asp58 (the site of phosphorylation) abolished all phosphorylation-dependent activities of RssB. While the phospho-mimic mutant D58E enabled a “strong” interaction with σ^s^ (~75% of RssB~P) this binding was no longer activated by the addition of AcP (Figure 6a, red bars). In contrast, replacement of Asp58 with Lys, not only resulted in a significant reduction to σ^s^ binding but also resulted in a loss of activation by phosphorylation (Figure 6a, orange bars). Consistent with a loss of binding to σ^s^ by both mutant proteins, there was a corresponding loss in the RssB-mediated degradation of the substrate by ClpXP (Figure 6d). Taken together these data suggest that phosphorylation of Asp58 stabilizes a “substrate-binding” conformation of RssB.

Next, to examine how the phosphorylation of Asp58 is signaled to the rest of the protein, we mutated Lys108 (which forms a H-bond with Asp58) (Figure 6c). Consistent with mutation of Asp58, replacement of Lys108 (with either arginine, aspartate or alanine) not only reduced σ^s^ interaction significantly, but also abolished the phosphorylation dependent activity of RssB (Figure 6a, purple bars, data not shown). These data suggest that Lys108 is, not only, critical for the formation of the “activated” complex—that recognizes σ^s^ with high affinity, but it also plays a critical role in sensing Asp58 phosphorylation. Finally, we speculated, that the signaling of Asp58 phosphorylation would be transferred to the C-terminal via an extended helix composed of α5 (from RssB_N_) and α6 (from RssB_C_).

Serendipitously, in our search for XBR-like motifs in RssB, we had already generated a mutation with the ^116^LREM^119^ motif located in the α5 helix. To examine the role of this region in σ^s^ turnover, we replaced the charged residues (RE), central to this motif, with alanine (here termed RE/AA). Initially we examined the ability of wild type or mutant RssB, to bind σ^s^ and deliver it to ClpXP for degradation, in the presence and absence of AcP. Interestingly, although σ^s^ recognition by this mutant was completely abolished in the absence of AcP, substrate interaction (and degradation) was partially restored in the presence of AcP (Figure 6b and Appendix A). These data suggest that rather than playing a direct role in σ^s^ binding, the RE motif appears to play an important role in stabilizing the “substrate-binding” conformation of RssB. One interpretation of these data is that the RE motif guides RssB towards the substrate “engaged” state, however activation to the “high affinity” conformation still requires phosphorylation. Importantly, activation by phosphorylation can overcome this defect in σ^s^ binding. Next, to further dissected this motif, we generated a single point mutant in which Arg117 was replaced with Ala (RssB_R117A_) here termed R117A. Consistent with the RE/AA mutant, substrate recognition by R117A was again absolutely dependent on phosphorylation (Figure 6a, blue bars). Taken together these data suggest that Arg117 is a critical “switch” residue required for σ^s^ recognition, in the absence of phosphorylation. We speculate that this residue forms an important element in the formation of an “engaged” conformation, possibly via a key interdomain interaction. Notably, the R117A mutant retained the ability to “deliver” σ^s^ to ClpXP, both in the presence and absence of AcP (Figure 6e, squares) despite no observable interaction with σ^s^ in the absence of AcP (as determined by pull-down, see Figure 6a,b). These data suggest that (i) σ^s^ preparation only requires a transient interaction with RssB and (ii) activation of ClpX (by RssB) is required for σ^s^ turnover, and (iii) R117A retains the ability to interact with and activate ClpX for σ^s^ turnover.

Next, in order to better understand the interaction between σ^s^ and RssB we performed small-angle X-ray scattering (SAXS). Given the apparent conformational flexibility of RssB, we used the R117A mutant (in the presence of AcP) to limit different conformations of RssB. Preliminary SAXS experiments of RssB alone indicated the presence of high MW scattering particles, consistent with aggregated proteins. To remove these contaminating aggregated proteins, which interfered with the SAXS analysis, samples were fractionated by SEC in-line with SAXS data collection. The SEC elution profile (Figure 7a), of Trx-RssB and σ^s^ alone indicated that aggregated protein (Ve ~52 mL, Figure 7a) was well separated from monomeric Trx-RssB and monomeric σ^s^, both of which eluted with a volume of ~90 mL (Figure 7a, red and blue lines respectively). As expected, the Trx-RssB/σ^s^ complex eluted earlier (Ve ~82 mL) than either protein alone (Figure 7a, black line) and importantly was clearly separated from the aggregated protein peak. The SAXS data obtained for Trx-RssB, σ^s^ and the Trx-RssB/σ^s^ complex (Figure 7b, red, blue and black lines, respectively) were analyzed to reveal information about MW and shape (Appendix A). Based on an estimate of the volume of the scattering particle, termed Porod analysis, the MW of Trx-RssB_R117A_ was determined to be 49 kDa. This MW was similar to the theoretical mass of Trx-RssB, (determined from its amino acid sequence; 52 kDa), and hence is consistent with Trx-RssB forming a well-structured protein composed of globular domains. In contrast, the MW for σ^s^ (as determined by analysis of the SAXS data) was 45 kDa, which is higher than the theoretical MW of σ^s^ (38 kDa). This could indicate some self-association and/or a higher apparent volume due to significant conformational averaging. Both Trx-RssB and σ^s^ had similar maximum dimensions (~105 Å) and very similar *R_g_* values (~31 Å), consistent with both proteins being elongated and composed of multiple domains. The SAXS analysis of the Trx-RssB/σ^s^ complex indicated a MW of ~91 kDa (in comparison to the theoretical MW of 92 kDa), which is also in good agreement with the sum of the SAXS-derived mass of each component (45 + 49 = 94 kDa). It also suggests that the factors that caused a high apparent mass for σ^s^ (self-association and/or conformational averaging) may persist in the complex. However, the *Dmax* of the Trx-RssB/σ^s^ complex is only ~25 Å longer than either of the individual components, suggesting these elongate components bind in a somewhat more globular arrangement.

Ab initio shape reconstruction of the SAXS data supports this with both isolated components being relatively narrow and elongate (Figure 7c,d), while the complex is considerably wider at one end (Figure 7e). An atomic resolution structure is not available for σ^s^, however based on homology to other *E. coli* sigma factors (free or in complex with RNA polymerase; PDB: 1SIG, 1L9U, 1KU2 and 3IYD), σ^s^ is expected to be highly flexible and likely to change its structure upon complex formation. Model building using rigid body refinement of linked structural fragments against the SAXS data was attempted, but in the absence of an experimental structure for σ^s^ and with long inter-domain linkers, no consistent solutions emerged. Thus, the X-ray crystal structures of RssB_N_ and RssB_C_ (see Appendix A) together with the structure of Trx, were manually docked into the Trx-RssB envelope, using the relatively small Trx domain as a reference point. Trx was used to orient the organisation of the complex and positioned in the narrow protrusion at one end of the envelope (RssB_N_ and RssB_C_ were too large to occupy this site), RssB_N_ was placed proximal to Trx, and RssB_C_ was positioned in the remaining distal unoccupied electron density (Figure 7c). Assuming Trx-RssB maintains a similar domain arrangement in the complex, comparison of the ab initio shape envelope of the Trx-RssB/σ^s^ complex with free Trx-RssB was performed (Figure 7e). This suggests that σ^s^ may wrap around the end of Trx-RssB, distal to Trx, making contact with both the N- and C-terminal domains of RssB. While other orientations may be possible, this arrangement, where σ^s^ binding seems to be mediated predominately through the C-terminal domain of RssB, is consistent with in vitro pull-down and GA cross-linking experiments.

Finally, we examined the mechanism of delivery of σ^s^ (by RssB) to ClpX(P). Previously, it was proposed that the N-terminal region of σ^s^ contains a ClpX-docking motif [9,48] and exposure of this motif (by RssB) was required for delivery of σ^s^ to ClpX. To confirm this possibility, we generated a deletion mutant of σ^s^ which lacked the first 50 residues of σ^s^. Initially, we tested the ability of RssB to deliver wild type or mutant σ^s^ to ClpX for degradation by ClpP (Figure 8a). Consistent with the findings of Hengge and colleagues [48], deletion of the first 50 residues of σ^s^ abolished the RssB-mediated turnover of σ^s^ (Figure 8a, compare lanes 1–5 with lanes 6–10). Next to ensure that removal of 50 residues from the N-terminus of σ^s^ did not compromise RssB interaction, we examined the ability of ΔNσ^S^ to compete with the RssB-mediated turnover of wild type σ^S^. Importantly, the RssB-mediated turnover of σ^S^ was inhibited in the presence of equimolar concentrations of ΔNσ^S^ (Figure 8a, blue squares). This turnover was further inhibited in the presence of a 2-fold (Figure 8a, red triangles) and 5-fold (Figure 8a, green diamonds) excess of ΔNσ^S^. Indeed, in the presence of a 5-fold excess of ΔNσ^S^ the half-life of σ^S^ increased from ~2.5 min to ~ 20 min. In order to confirm that ΔNσ^S^ retained the ability to interact with RssB we performed a series of pull-down experiments (Figure 8b). Consistent with the substrate competition experiments, equal amount of wild type or ΔN σ^S^ were recovered from immobilized RssB (Figure 8b, compare lanes 4 and 7, respectively). Collectively, these data suggest that RssB is required for the release of the N-terminal ClpX binding motif, which is recognized by ClpX.

## 4. Discussion

The general stress response is a crucial cross-protective stress response pathway. In *E. coli* the pathway is controlled by a specialized sigma factor, σ^S^. Given the importance of this pathway it is not surprising that the levels of σ^S^ are highly regulated, not only at the transcriptional and translational level but also post-translational level. Consistent with the stringent control of σ^S^ levels in the cell, the turnover of σ^S^ is strictly regulated by many different cellular components, including ATP levels and several different proteins (adaptors and anti-adaptors) [22,24,25]. A key regulator of the pathway, that is central to the turnover of σ^S^, is the adaptor protein RssB. This adaptor protein is a member of the RR family and as such is composed of two domains, an N-terminal REC domain that can be phosphorylated on a conserved Asp residue (Asp58) and a C-terminal effector domain. However, in contrast to most RRs (in which the effector domain is involved in DNA binding) RssB is an atypical member of the RR family as the effector domain is involved in protein binding. Indeed, the effector domain of RssB exhibits a PPM/PP2C phosphatase fold, however it lacks the active site residues. For these reasons, the mechanism of action of RssB has remained poorly understood [49]. Here we report the functional characterization of RssB, using a range of biochemical (pull-downs, immunoprecipitation and chemical crosslinking) and structural approaches. Our data help define the mechanism of action of this intriguing adaptor protein, how it interacts with, prepares, and delivers its substrate (σ^S^) to the unfoldase (ClpX) for degradation by ClpP. Based on our data, together with published findings we propose a model for the engagement of RssB with its substrate σ^S^, its docking to ClpX and the delivery of σ^S^ to the ClpXP protease.

Similar to other AAA+ adaptor proteins, RssB docks to its cognate unfoldase component (ClpX) via a specific interaction with the N-terminal ZBD [12,13,15,50,51]. The C-terminal effector domain of RssB is essential for this docking (Figure 2). However, in contrast to our previous model [12], the interaction (between RssB and the ZBD) is mediated, not via the C-terminal sequence of RssB (as it is critical to the fold of RssB) but rather by an alternative XBR-like region within the C-terminal domain (Figure 1). Based on our peptide library analysis we propose that the ZBD of ClpX interacts with the α10 helix of RssB (Figure 2). Interestingly, this region was also recently identified as part of the IraD binding site in RssB [23]. Collectively, this suggests that both the ZBD of ClpX and IraD share a common or overlapping docking site within RssB. Hence, we propose that docking of IraD to this site is likely, not only, to prevent σ^S^ binding to RssB, but also prevent the delivery of σ^S^ to ClpX(P). Interestingly, in addition to its interaction with IraD and ClpX, the C-terminal domain of RssB appears to bind to all components of the pathway, as it also makes direct contact to the anti-adaptors, IraM and IraP, [22,23,24] as well as the substrate, σ^S^ (Figure 5). This suggests that the C-terminal domain of RssB represents a “hub” for the competitive docking of most, if not all, factors involved in σ^S^ turnover (Figure 9). This, competitive binding to a common or overlapping platform in RssB, is an efficient way to ensure that σ^S^ turnover can be rapidly and reversibly controlled.

Although the C-terminal domain of RssB serves as a common site for docking of each of the different components involved in the turnover of the σ^S^, the interaction of many of these components is modulated by the N-terminal domain. In some cases, the interaction (e.g., with σ^S^) is enhanced by phosphorylation. Although the physiological significance of RssB phosphorylation remains controversial, our in vitro data suggests that similar to other RRs [49], the phosphorylation of RssB enhances its biological activity, as it increases both the binding and turnover of σ^S^ by ~3-fold. Based on our mutational analysis of conserved residues located within the N-domain, we propose that phosphorylation of RssB (at Asp58) is sensed by the conserved residue (Lys108) located within the 4-5-5 interface. This interaction stabilizes the α5 helix of RssB in an “activated” conformation which facilitates high affinity recognition of σ^s^. Interestingly, consistent with an important biological role for this region, our structural comparisons of RssB with several other RRs, including the recent structure of the RssB-IraD complex (r.m.s.d. of 1.4 Å), demonstrated that this region is mobile as the α5 helix was observed to adopt several different conformations (Appendix A). Similarly, this region displayed high B-factors in our structure of the REC domain. Consistent with an important role for the α5 helix, we also discovered that, in the absence of phosphorylation, a single residue (Arg117, located on the distal end of α5 helix) was essential for substrate interaction (Figure 6). Importantly, although this residue is essential for σ^S^ binding in the absence of phosphorylation, the interaction with substrate was restored in the presence of AcP (Figure 6). These data clearly demonstrate that Arg117 is not directly involved in substrate interaction, but rather promotes or stabilizes a conformation of RssB that supports substrate interaction. Interestingly, movement of the α5 helix (in the RssB-IraD complex) places Arg117 at the interface of both RssB domains and IraD, where it forms a H-bond with Gln247 of the C-domain [23]. Therefore, we speculate that Arg117 represents a “switch” residue in RssB, that depending on the position of the α5 helix, either stabilizes RssB in an “activated” state or an “inhibited” state.

Adaptor proteins are specific recognition factors that regulate the delivery of a particular substrate (or class of substrates) to an individual AAA+ protease [11,13,14,52]. Overall, these proteins can be classified into two groups, based on their mechanism of action. The first group are scaffold adaptors, which tether the bound substrate to the AAA+ unfoldase. These proteins (e.g., the ClpX adaptor SspB, involved in the turnover of SsrA-tagged substrates) generally enhance the rate of substrate turnover for a substrate that is intrinsically recognized by its cognate AAA+ unfoldase. Typically, this type of adaptor is composed of two well-defined regions—a substrate binding domain and a protease docking region. The second group of adaptors are “activating” adaptors. In contrast to the “tethering” adaptors, these proteins (e.g., ClpS, MecA, and RssB) are essential for the turnover of one or more substrates, and as a result the mechanism of action is generally more complex. Some “activating” adaptors (such as MecA and McsB) control AAA+ protease assembly and hence are generally required for most proteolytic activities of that machine [53,54], while other “activating” adaptors are only required for the delivery (and turnover) of a specific substrate (or substrate class). In the second group of “activating” adaptors, the adaptor protein may be required either for substrate activation or substrate delivery or in some cases both activities. One of the best characterized examples of an “activating” adaptor is the N-recognin, ClpS, which is essential, not only, for N-degron recognition but also for the substrates’ subsequent delivery to ClpA [55,56,57,58]. In this case, the delivery process can be separated into several important steps, from low affinity docking to the N-terminal domain of the AAA+ protein, activation of the unfoldase and ending in active substrate release. Similar to ClpS, we propose that σ^S^ delivery by RssB involves a number of discrete steps. In the first step, RssB interacts with its substrate σ^S^. This interaction is primarily mediated by direct contact to the C-terminal domain of RssB, however docking to this domain can be modulated in the presence of the N-terminal domain, which can be enhanced by phosphorylation. In the absence of phosphorylation, a specific residue, R117 (located on the α5 helix between the two domains) controls substrate “engagement”. We speculate that R117 plays a key role in stabilizing different RssB conformations. Finally, after engagement with the substrate, it’s been proposed that the N-terminal region of σ^S^ is “revealed” for delivery to ClpX. In addition to this “preparation” step, we speculate that RssB is also require for activation of ClpX. This is based on the identification of several RssB mutants that fail to bind σ^S^ but yet still facilitate σ^S^ turnover (Figure 6).

## 5. Conclusions

In contrast to several bacterial adaptor proteins, the mechanism of action of RssB has so far proved elusive. Progress in this field has been largely hampered by the atypical domain structure of this RR, the weak affinity of its individual components (for their binding partners) and until recently, by a lack of structural information for any of the proteins involved in the turnover of σ^S^. To overcome the weak affinity of the various interactions, we have used chemical crosslinking to help define the molecular steps involved in the recognition of σ^S^ by RssB and its delivery to ClpX. Importantly, we have been able to validate each of these interactions using a number of different approaches. In addition to describing the broad details of these interactions, we have also identified several important residues within the N-terminal REC domain of RssB that are required for the phosphorylation dependent recognition of its substrate. We have also identified a single residue (R117), that is essential for σ^S^ recognition (in the absence of phosphorylation), but expendable in the presence of phosphorylation. Significantly, R117 is located on a mobile helix, the α5 helix, that links the two domains, hence we speculate this residue plays a significant role in communication between the two domains. Finally, we have developed a preliminary SAXS model of RssB in complex with its substrate σ^S^. Based on these data, we have developed a preliminary model for the mechanism of action of RssB which we hope will stimulate future efforts to study this novel adaptor protein. Clearly, further dissection of the delivery mechanism is still required and additional biochemical and structural experiments are eagerly awaited.

## Figures and Tables

**Figure 1 biomolecules-10-00615-f001:**
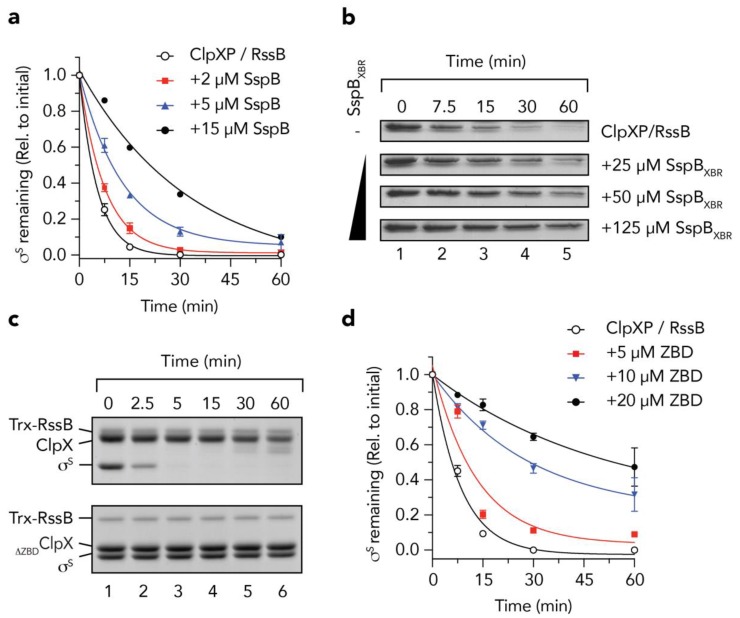
The zinc binding domain (ZBD) of ClpX is an essential platform for the RssB-mediated degradation of σ^s^. (**a**) The RssB-mediated turnover of σ^s^, by 1.2 µM ClpXP (open circles) is inhibited by the addition of 2 µM (red squares), 5 µM (blue triangles) and 15 µM (black circles) SspB. Three independent experiments were performed (n = 3), except for 15 µM SspB, where a single experiment was performed. Error bars represent the standards error of the mean (S.E.M.). (**b**) The RssB-mediated turnover of σ^s^, by ClpXP was monitored in the presence of increasing concentrations of SspB_XBR_. (**c**) The RssB-mediated degradation of σ^s^ (by ClpP) was monitored in the presence of either ClpX (upper panel) or ClpX lacking the ZBD (_∆ZBD_ClpX) (lower panel). (**d**) The RssB-mediated turnover of σ^s^, by 0.6 µM ClpXP (open circles) is inhibited by the addition of 5 µM (red squares), 10 µM (blue triangles) and 20 µM (black circles) ClpX_ZBD_ (ZBD). Error bars represent standard error of the mean (S.E.M.) from 3 independent experiments (n = 3).

**Figure 2 biomolecules-10-00615-f002:**
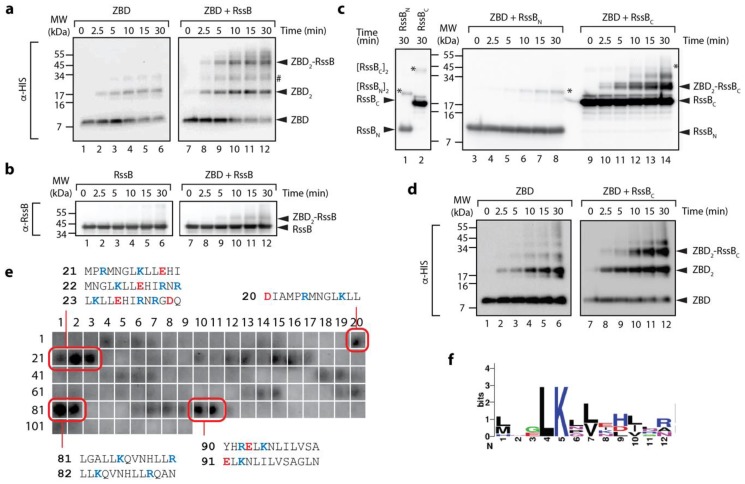
The ZBD of ClpX interacts with C-terminal domain of RssB. (**a**,**b**) The interaction of the ZBD of ClpX with itself (lanes 1–6) or RssB (lanes 7–12) was examined by chemical crosslinking using 0.005% (*v/v*) glutaraldehyde (GA). Following crosslinking, proteins were separated by SDS-PAGE, transferred to PVDF membrane and probed with (**a**) anti-His antisera (for ZBD) or (**b**) anti-RssB (for RssB). In the absence of any additional proteins, the ZBD of ClpX assembled into dimers (lanes 1–6). Following the addition of RssB to ZBD (in the presence of GA) a specific crosslinked protein band (ZBD-RssB) of ~45 kDa was observed by immunodecoration using (**a**) anti-His antisera and (**b**) anti-RssB antisera. # represents a non-specific crosslink of 2 ZBD dimers. (**c**) The ZBD of ClpX forms a specific crosslink with RssB_C_. Following crosslinking, proteins were separated by SDS-PAGE, transferred to PVDF membrane and probed with anti-RssB antisera. In the absence of the ZBD, RssB_N_ (lane 1) and RssB_C_ (lane 2) form dimers (indicated by *). In the presence of the ZBD of ClpX, a specific crosslink to RssB_C_ (lanes 9–14) and not to RssB_N_ (lanes 3–8) was observed. (**d**) The ZBD of ClpX forms a specific crosslink with RssB_C_. To confirm the presence of ZBD, reactions were probed with anti-His antisera. In the absence of RssB_C_, the ZBD formed dimers (lanes 1–6). Following addition of RssB_C_ an additional crosslinked protein band (ZBD_2_-RssB_C_) was observed (lanes 9–12). (**e**) The ZBD interacts with 13-mer peptides in RssB (numbering refers to peptide spots, sequences of “strong” binding peptides are indicated). (**f**) Weblogo representation of the consensus motif for “strong” binding peptides to the ZBD of ClpX.

**Figure 3 biomolecules-10-00615-f003:**
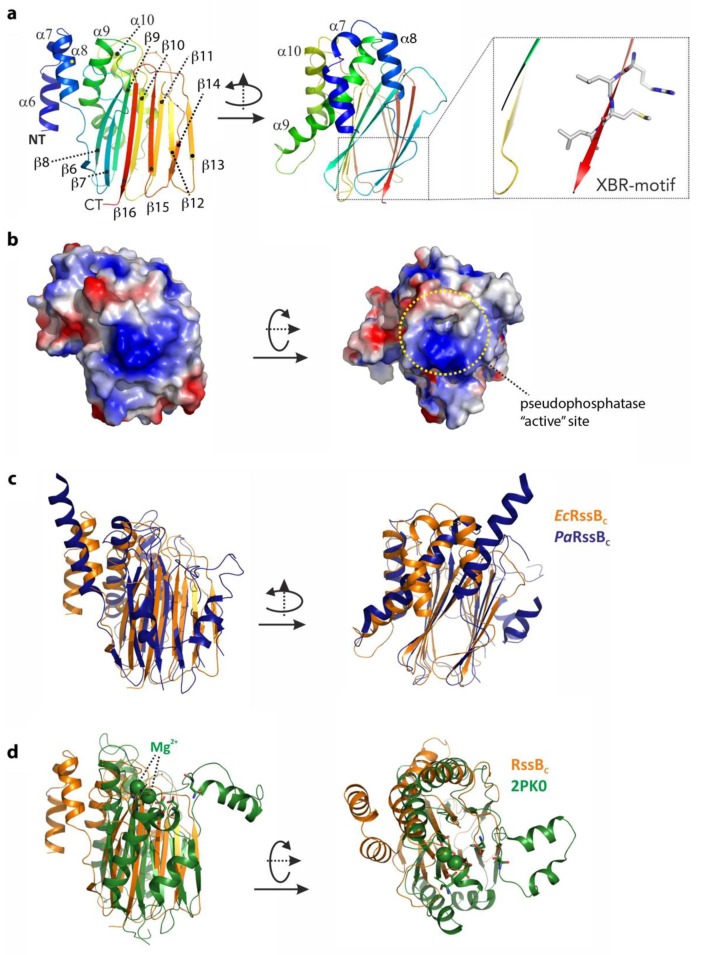
The structure of the C-terminal domain of RssB (RssB_C_) and superposition with close structural homologs. (**a**) The structure of the C-terminal domain of RssB is shown in cartoon representation from two perspectives related by a rotation of 90 degrees around the *Y*-axis. The structure is a mixed α/β-structure formed by α-helices α6–α10 and β-sheets β6–β16. The C-terminal XBR-motif is shown (boxed). (**b**) Surface representation of the C-terminal domain with the electrostatic potentials marked. The left panel shows the same orientation used in (**a**), the right orientation was rotated by 90 degrees around the *x*-axis. The putative active site is marked. (**c**) Structure comparison of RssB_C_ shown in the same orientation as in (**a**) with the C-terminal domain of RssB from *P. aeruginosa* displayed in dark blue. (**d**) Superposition of RssB_C_ shown in orange (same orientation as is (**a**)) with the serine/threonine phosphatase structure of *S. agalacticae* shown in green (PDB-entry 2PK0). Magnesium atoms known to be important for function of the phosphatase are marked in ball representation and active site residues are shown in stick representation.

**Figure 4 biomolecules-10-00615-f004:**
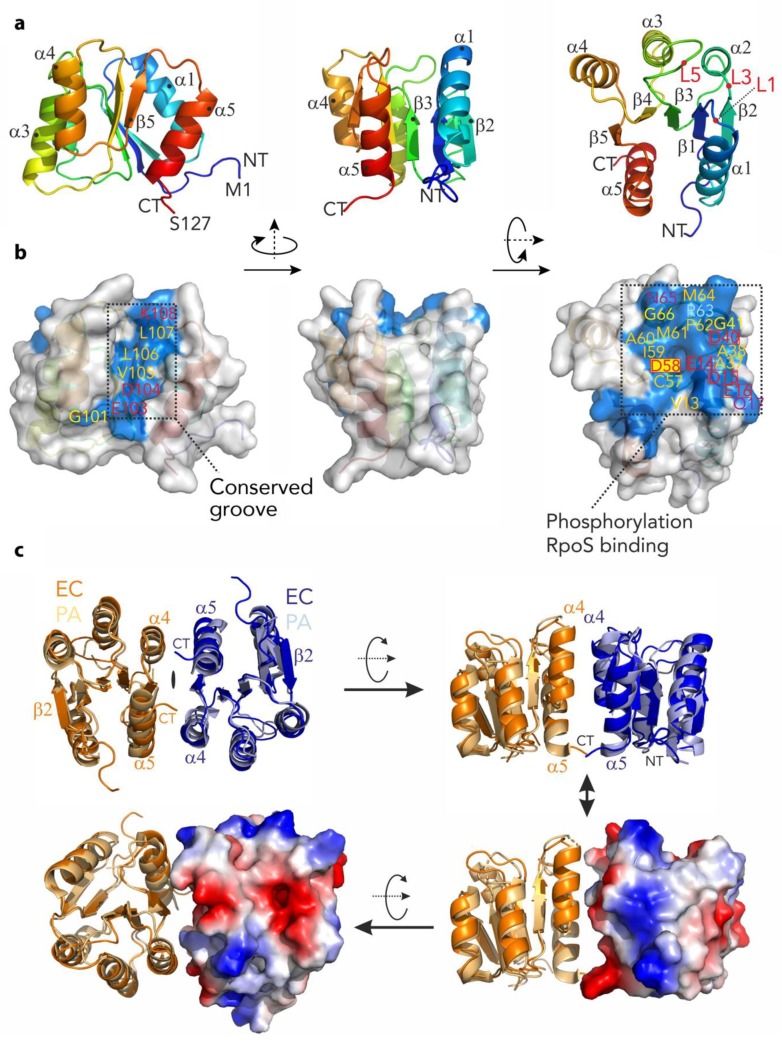
The structure of the N-terminal domain of RssB (RssB_N_) and putative dimer. (**a**) Structure of the N-terminal domain shown in cartoon representation and color coded from the N- (NT; Met1; in blue) to the C-terminus (CT; Ser127; in red) using rainbow colors. The structure is shown from three different angles which are related by rotation of 90 degrees around the *y*-axis and second rotation of 90 degrees around *x*-axis. The five-stranded β-sheet (β1–β5) is surrounded by α-helices (α1–α5). (**b**) Structure of RssB_N_ in the same orientation as in (**a**) as surface representation with conserved residues colored in blue and named according to the sequence. Hydrophobic residues are marked in orange, glycine residues are marked in yellow, hydrophilic residues are marked in magenta and charged residues are marked in red. The conserved phosphorylation site on Asp58 is marked with a yellow rectangle. There are two conserved patches visible on the surface. The conserved groove on the left represents a putative dimerization interface, while the extended and conserved surface on the right is indicative for interactions with the substrate protein σ^s^ after phosphorylation. This patch is particularly negatively charged with a high density of acidic residues. (**c**) Comparison of the N-terminal domain structures of RssB from *E. coli* (EC—in dark orange and dark blue) and *P. aeruginosa* (PA—in light orange and light blue). Three different perspectives are shown. The structures are virtually identical for the β–sheet and adjacent α–helices but varies in loop structures (superimposed with a r.m.s.d. of 1.3 Å). In the lower images the electrostatic surface of one of the subunits from the *E. coli* protein is illustrated.

**Figure 5 biomolecules-10-00615-f005:**
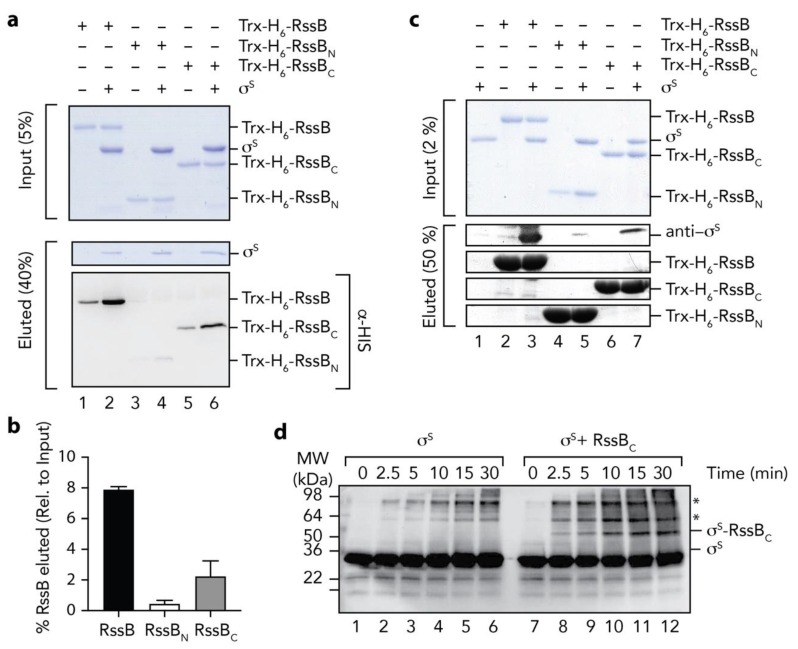
The C-terminal domain of RssB is sufficient for interaction with σ^s^. (**a**) RssB (lanes 1 and 2), RssB_N_ (lanes 3 and 4), or RssB_C_ (lanes 5 and 6) were incubated in the absence (lanes 1, 3, and 5) or presence (lanes 2, 4 and 6) of σ^s^. RssB (wild type or individual domains) were co-immunoprecipitated using anti-σ^s^ antisera. Proteins were separated by 15% SDS-PAGE and visualized by either staining with Coomassie Brilliant Blue or transfer to PVDF membrane and immunodecoration with anti-His (to visualize Trx-H_6_-RssB). The input (5%) is shown in the upper panel and the eluted proteins (40%) are shown in the lower panels. (**b**) Quantitation of in vitro σ^s^ co-IP experiments show the binding of σ^s^ to Trx-H_6_-RssB_N_ (white bar) and Trx-H_6_-RssB_C_ (grey bar), relative to wild type Trx-H_6_-RssB (black bar). Error bars represent S.E.M. of at least three independent experiments (n > 3). (**c**) In vitro σ^s^ “pull-down” experiment illustrating the interaction of σ^s^ (lane 1) with wild type Trx-H_6_-RssB (lanes 2 and 3), Trx-RssB_N_ (lanes 4 and 5) and Trx-RssB_C_ (lanes 6 and 7), in the presence (lanes 3, 5 and 7) or absence (lanes 2, 4, and 6) of σ^s^. Proteins were separated by 15% SDS-PAGE and visualized by staining with Coomassie Brilliant Blue or immunodecoration using anti-σ^s^ antisera. The input (2%) is shown in the upper panel and the eluted proteins (50%) are shown in the lower panels. (**d**) Chemical crosslinking of σ^s^ (using 0.005% (*v/v*) GA) was monitored in the absence (lanes 1–6) or presence (lanes 7–12) of RssB_C_. Following crosslinking, proteins were separated by SDS-PAGE, transferred to PVDF membrane and probed with anti-σ^s^ antisera. * represents non-specific σ^s^ crosslink products.

**Figure 6 biomolecules-10-00615-f006:**
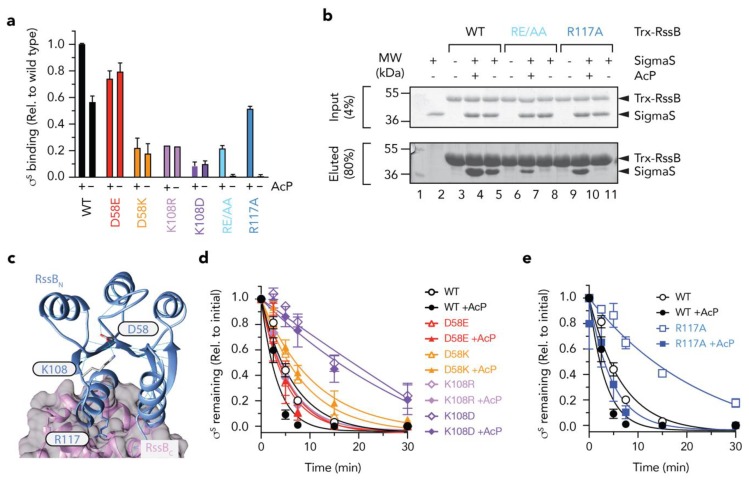
The N-terminal domain of RssB regulates docking to σ^s^. (**a**) Quantitation of in vitro σ^s^ “pull-down” experiments showing the binding of σ^s^ to various RssB mutant proteins (D58E, red bars; D58K, orange bars; K108R, mauve bars; K108D, purple bars; RE/AA, light blue bars; R117A, dark blue bars) relative to wild type Trx-RssB (black bars) in the presence or absence of AcP. (**b**) In vitro σ^s^ “pull-down” experiment illustrating the interaction of σ^s^ (lane 2) with wild type Trx-RssB (lanes 3–5), Trx-RssB_RE/AA_ (lanes 6–8) and Trx-RssB_R117A_ (lanes 9–11), in the presence (lanes 4, 7, 10) or absence (lanes 5, 8, and 11) of AcP. MW markers (lane 1). Proteins were separated by 15% SDS-PAGE and visualized by staining with Coomassie Brilliant Blue. The input (4%) is shown in the upper panel and the eluted proteins (80%) are shown in the lower panel. (**c**) Structure of RssB (in IraD bound conformation; PDB: 6OD1) illustrating location of D58, K108 and R117A. (**d**,**e**) Quantitation of ClpXP-mediated degradation of σ^s^ by wild type and mutant RssB in the presence or absence of AcP (as described in (b). Error bars represent S.E.M. of at least three independent experiments (n > 3), except in the case of K108R, in which only a single experiment was performed (n = 1) and hence no error bars are shown.

**Figure 7 biomolecules-10-00615-f007:**
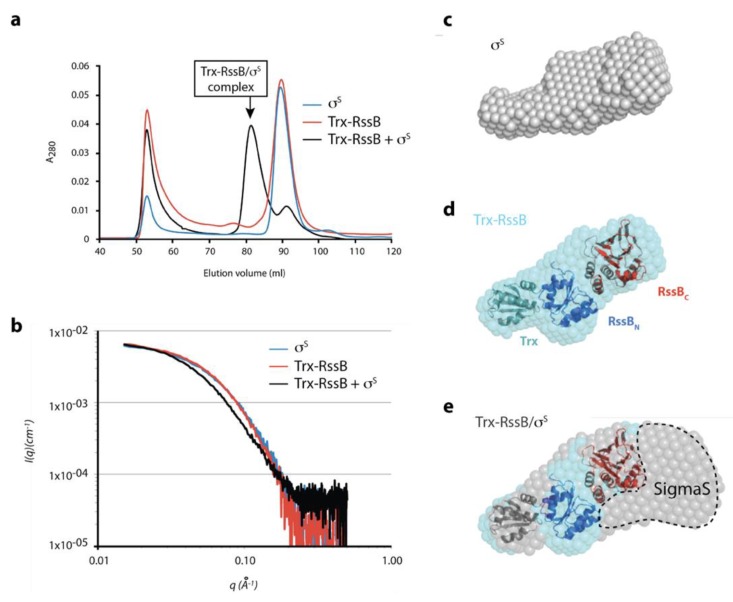
Preliminary SAXS model of the RssB/σ^s^ complex (**a**) SEC elution profile of σ^s^ (blue line), Trx-RssB (red line) and Trx-RssB in complex with σ^s^ (black line). (**b**) Average small angle X-ray scattering (SAXS) data for monomeric σ^s^ (blue line), Trx-RssB (red line), and Trx-RssB in complex with σ^s^ (black line). The intensities are represented as a function of the magnitude of the scattering vector (*I(q)* vs. *q*). (**c**) the SAXS-derived average ab initio shape envelope for σ^s^ (grey spheres), (**d**) the SAXS-derived average ab initio shape envelope for Trx-RssB (cyan spheres) with Trx-RssB domains: Trx (green), RssB_N_ (blue) and RssB_C_ (red) manually docked into the envelope. Trx was placed into the small protrusion, the N-domain adjacent to it and C-domain distal to Trx. (**e**) The SAXS-derived average ab initio shape envelope for Trx-RssB in complex with σ^s^ (grey spheres), overlaid with the shape envelope of Trx-RssB (cyan spheres) showing individual domains (Trx, RssB_N_, and RssB_C_) from (**d**), highlighting the location of σ^s^ (SigmaS).

**Figure 8 biomolecules-10-00615-f008:**
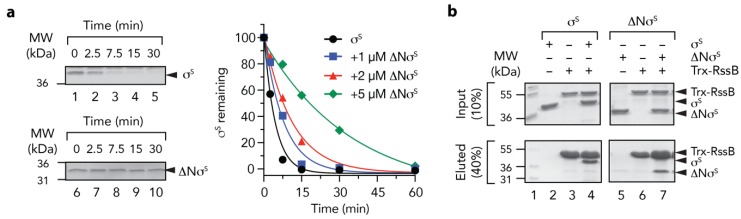
The N-terminus of σ^s^ is required for delivery to ClpX(P) (**a**) The RssB-mediated turnover of σ^s^, by 0.6 µM ClpXP (lanes 1–5) is inhibited by removal of the N-terminus (residues 1 - 50) of σ^s^ (∆Nσ^s^) (lanes 6–10). Increasing concentrations of ∆Nσ^s^ [1 µM (blue squares), 2 µM (red triangles) or 5 µM (green diamonds) inhibit the turnover of wild type σ^s^]. (**b**) In vitro σ^s^ “pull-down” experiment illustrating the interaction of wild type σ^s^ (lanes 2–4) or ∆Nσ^s^ (lanes 5–7) with Trx-RssB (lanes 4 and 7) in the presence of AcP. MW markers (lane 1). Proteins were separated by 15% SDS-PAGE and visualized by staining with Coomassie Brilliant Blue. The input (10%) is shown in the upper panel and the eluted proteins (40%) are shown in the lower panel.

**Figure 9 biomolecules-10-00615-f009:**
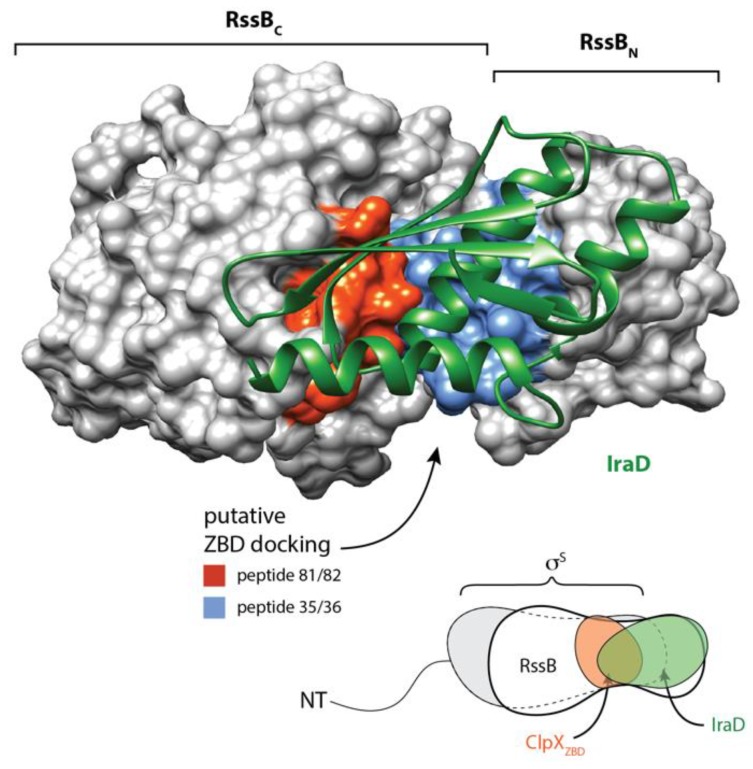
Structure of RssB-IraD complex (6OD1) highlighting putative ZBD interaction sites. IraD (green) is shown in ribbon representation and RssB is shown in surface representation. Putative surface exposed ZBD-interacting sites (as determined from peptide library binding) are shown in red (peptides 81 and 82) and blue (peptides 35 and 36).

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
