# Peer review of "Insight into the RssB-Mediated Recognition and Delivery of σ^s^ to the AAA+ Protease, ClpXP"

_biomolecules, 2020, doi:10.3390/biom10040615_

Round 1
Reviewer 1 Report
Evaluation of the manuscript entitled “Insight into the RssB-mediated recognition and delivery of SigmaS to the AAA+ protease, ClpXP” by Micevski et al.,
Micevski et al., address the delivery of Escherichia coli Sigma factor S to the ClpXP protease for subsequent degradation. Removal of this sigma factor represents a crucial step in stress response regulation in bacteria and is still not fully understood. The paper therefore addresses an important topic in bacterial physiology.
The delivery of SigmaS to the protease is performed by the adaptor protein RssB which mediates interaction between the sigma factor and the ClpX AAA-ATPase unfolding component. The authors used a wide variety of structural and biochemical experiments to investigate the RssB mediated transfer of SigmaS to ClpXP. These include determination of the crystal structures of the RssB N- and C-terminal domains as well as preliminary SAXS analysis of full length RssB. Extensive in vitro binding and degradation studies show that interaction requires the N-terminal domain of SigmaS and the C-terminal domain of RssB. In addition, phosphorylation at residue D58 in a conserved groove of the RssB N-terminal receiver domain is essential for high affinity interaction and modulates degradation of the sigma factor. Moreover, the authors show a key role of R117 that is important for N- and C- domain interaction in degradation, possibly by stabilizing the Sigma factor engaged conformation. The C-domain of RssB not only plays an important role in sigma factor interaction, but as shown by a peptide array, also in recognition of the ClpX ZBD domain which serves as docking platform for the SigmaS/RssB complex. The interaction with the ZBD domain was proposed to liberate the sigma factor after its deposition at ClpX and feeding its N-terminal portion into its pore. Interestingly, the same region interacting with the ZBD domain on RssB seems to be recognized by IraD suggesting that this anti-adaptor protein also competes for binding with ClpX.
The presented structural and biochemical data are of very high quality and the results are in line with the conclusions of the authors. Moreover, the manuscript is well written and the figures are well designed and easy to understand. I only have some minor comments that would make the manuscript better accessible for readers from outside the field.
Line 33 to 36:
33 to the Zinc Binding Domain (ZBD) of ClpX. We propose that the ZBD docking-site in RssB overlaps with the IraD docking-site, and hence speculate that docking to ClpX may trigger release of its substrate through activation of a “closed” state, thereby coupling adaptor docking with substrate release
Most readers might not be familiar with what IraD does and it is not mentioned in the text. Therefore, it will be hard to understand the meaning of this paragraph. It should be explained in more detail and it has to be mentioned that IraD is an anti-adapter protein.
98 Consistently, our in vitro data shows that
Should read show
Line 99/100:
99 In addition, based on our mutant (R117A), we have identified Arg117 as a key “regulator” of ss interaction.
A single residue can hardly be a “regulator”: using the term “key residue” would fit better.
Lines 110 to 114:
110 release the substrate. This “competitive” docking mechanism would ensure the efficient and timely release of SigmaS into the hexameric pore of ClpX, upon binding. Hence, we propose that RssB docking to SigmaS is not only essential for substrate activation, but also required for docking to the ZBD of ClpX, which may regulate timely release of SigmaS and the activation of ClpX for SigmaS recognition and translocation into ClpP.
It is confusing that the first sentence addresses the docking mechanism of RssB to ZBD of ClpX, while in the next sentence RssB docking to SigmaS is dealt with. The authors should also think about alternative words to docking.
262: Initially, to validate the idea that RssB and SspB use a common docking platform (on ClpX) for delivery of its substrates to ClpXP, we performed a series of adaptor competitions experiments in which we monitored the RssB-mediated turnover of ss, in the absence or presence of SspB (Figure 1).
The rationale behind using the stringent starvation protein SspB should be better explained in one sentence
289: assays, in which the RssB-mediated turnover of ss (by wild type ClpXP) was monitored in the
Remove parentheses
Legend to Figure: 5: Asterisk in the figure should be explained
Table 1 and Table 2 could be transferred into the supplements to benefit readability.
Author Response
We thank all of the reviewers for their feedback on our manuscript. We have now addressed all of their concerns. A point-by-point response for each of their concerns (in italics) is included below.
Response to Referee #1
33 to the Zinc Binding Domain (ZBD) of ClpX. We propose that the ZBD docking‐site in RssB overlaps with the IraD docking‐site, and hence speculate that docking to ClpX may trigger release of its substrate through activation of a “closed” state, thereby coupling adaptor docking with substrate release.
Most readers might not be familiar with what IraD does and it is not mentioned in the text. Therefore, it will be hard to understand the meaning of this paragraph. It should be explained in more detail and it has to be mentioned that IraD is an anti‐adapter protein.
We have made substantial changes to the abstract to address the concerns of Reviewers #1 and #3.
98 Consistently, our in vitro data shows that
We have replaced shows with show (as suggested).
99 In addition, based on our mutant (R117A), we have identified Arg117 as a key “regulator” of sigS interaction.
We have replaced “regulator” with “residue that regulates SigS interaction”.
110 release the substrate. This “competitive” docking mechanism would ensure the efficient and timely release of SigmaS into the hexameric pore of ClpX, upon binding. Hence, we propose that RssB docking to SigmaS is not only essential for substrate activation, but also required for docking to the ZBD of ClpX, which may regulate timely release of SigmaS and the activation of ClpX for SigmaS recognition and translocation into ClpP.
It is confusing that the first sentence addresses the docking mechanism of RssB to ZBD of ClpX, while in the next sentence RssB docking to SigmaS is dealt with. The authors should also think about alternative words to docking.
We have made several changes to this paragraph to remove confusion about the meaning of this aspect of the work.
262: Initially, to validate the idea that RssB and SspB use a common docking platform (on ClpX) for delivery of its substrates to ClpXP, we performed a series of adaptor competitions experiments in which we monitored the RssB‐mediated turnover of ss, in the absence or presence of SspB (Figure 1). The rationale behind using the stringent starvation protein SspB should be better explained in one Sentence
A brief rationale for using SspB/SspB(XBR) is described on line 90-93.
289: assays, in which the RssB‐mediated turnover of ss (by wild type ClpXP) was monitored in the
Parentheses have been removed, as suggested.
Legend to Figure: 5: Asterisk in the figure should be explained
An explanation for the asterisk has been added to the figure legend
Table 1 and Table 2 could be transferred into the supplements to benefit readability.
We have moved the tables (as suggested) to the supplementary section
Reviewer 2 Report
The manuscript describes a study of the interaction of the ClpXP adaptor protein RssB with both the protease and its binding partner, δs. The precise mechanism whereby RssB delivers δs for degradation by ClpXP is unclear and is an worthy area of study. The authors report that the C-terminal domain of RssB is primarily responsible for binding to both the zinc-binding domain of ClpX and to δs and that phosphorylation of the N-terminal domain of RssB increases the affinity of interaction with δs. The authors also report crystal structures of the isolated N- and C-terminal domains of RssB.
Major points:
An unpublished crystal structure of full-length Pseudomonas aeruginosa RssB has been deposited in the PDB. The authors only describe the rmsd values betwen these structures. Do residues similar to the RE motif in helix a5 exist in the P.aeruginosa structure? Does the conformation of the full-length protein help rationalize any of the arguments made by the authors based on their biochemical characterization.
Author Response
An unpublished crystal structure of full-length Pseudomonas aeruginosa RssB has been deposited in the PDB. The authors only describe the rmsd values betwen these structures. Do residues similar to the RE motif in helix a5 exist in the P.aeruginosa structure? Does the conformation of the full-length protein help rationalize any of the arguments made by the authors based on their biochemical characterization.
Although there is extensive evidence to demonstrate that E. coli RssB is directly involved in SigS turnover, unfortunately there is currently no direct evidence to suggest that Pseudomonas aeruginosa RssB regulates SigS activity, more specifically there is no evidence to suggest a role for P. aeruginosa RssB in the turnover of SigS. Consistent with this divergent role of RssB in P. aeruginosa, there are many differences in the sequence, overall structure and proposed mechanism of action of E. coli RssB and P. aeruginosa RssB. In contrast to E. coli RssB, the C-terminal phosphatase domain of P. aeruginosa RssB contains all of the active site residues (Galperin, et al., 2006). In addition, P. aeruginosa RssB contains an extended linker region (which forms a coiled-coil region in the P. aeruginosa RssB structure), which is lacking in E. coli RssB. Finally, P. aeruginosa RssB is found in an operon with a putative anti-sigma factor. Collectively these findings suggest that P. aeruginosa RssB, likely regulates SigS not via degradation but rather by an upstream kinase in a similar way to the control of SigS, by CsrR (RssB homolog) and CsrA in Shewanella.
Consistent with these proposed divergent roles of RssB in these two species, we do not see similar residues in the a5 helix (shown in new Supp figure S1). Moreover, because of these likely significant differences in the modes of action of the two proteins, we don’t believe it is appropriate to use the full-length structure of P. aeruginosa RssB to help to rationalise our proposals based on our biochemical analysis.
Reviewer 3 Report
In this manuscript, Micevski et al. provide a very detailed analysis of the interaction of the E. coli RssB adaptor with the substrate σS and the ATPase ClpX. They solved the X-ray crystal structures of RssB N- and C-terminal domains. They also described the SAXS structure of full length RssB. They map the interaction interfaces between RssB and the zinc binding domain (ZBD) of ClpX as well as between RssB and σS. Interestingly, they find that the docking site of the anti-adaptor IraD on RssB C-domain overlaps with that of ZBD. Furthermore, although the phosphorylation of the N-terminal domain (Asp58) was found to be essential for high affinity binding of σS, most of the direct binding occurs via the C-terminal domain of RssB. They identified the α5 helix of the N-terminal domain as being important for the formation of an engaged or low affinity conformation of RssB which drives interaction with σS at the distal C-terminal domain. The data is nicely presented and the conclusions are supported by the experiments.
I have a few comments.
- I would recommend rewording the abstract since it is difficult to follow for someone who is not in the field. They need to provide more background information in the abstract.
- The overlay of the current RssB structures with previous RssB structures (Figures S1 and S4) should be part of the main figures. This is important for the reader to follow the reasoning of the manuscript.
- Figure 1c, lanes 5 and 6 – Is ClpX being cleaved at 30 and 60 min? If so, this observation should be discussed. Error bars are not clear in Fig. 1d – make them black, for example.
- In Figure 2a, I am surprised that they see a monomer of ZBD bound to a monomer of RssB. ZBD is a very tight dimer (nM Kd), and I would expect the main species to be a dimer of ZBD and a monomer of RssB. The authors should re-evaluate their results.
- Figure 2d – The authors should explain how the peptides were generated and what is the overlap between the peptides.
- Line 353 – The authors state that no cross-linking was observed between ZBD and RssBN. This data should be shown.
- Lines 357 and 358. Label peptides 20 – 23, 35 – 36, 81 – 82, and 90 – 91 in Figure 2d so readers can assess the significance of these RssB peptides in binding to ZBD.
- The arrows in Figure 4a showing the directions of rotations do not seem to be correct. Also, label L2 in right most panel.
- In the experiments of Fig. 5, does phosphorylated RssBN (or using a phosphor mimic putant) bind σS?
- What does the star represent in Fig. 5d?
- It might be worth to check if the active and inactive conformation od RssB can be distinguished by CD.
- Line 571 – It is not clear why the authors mutated K108. A rationale is needed.
- Why didn’t the authors use their own structure in Figure 6c?
- Line 616 – what is the evidence that the R117A mutant reduces the conformational dynamics of RssB?
- Finally, do the authors have an idea of the binding site of RssBC on ZBD? Identifying such a site would further our understanding of the interactions formed by RssB leading to enhanced σS degradation.
Minor
- Line 55 – ‘resistance’ should be ‘stress’.
- Line 110 – ‘release the substrate’ should be ‘release of the substrate’.
- Line 261 – delete ‘but not via the XBR of RssB’ since XBR is not introduced yet.
- Line 577 – the section starting with ‘Serendipitously, in our search for alternate…’ should be a new paragraph
- A movie is listed as part of supplementary material. However, I did not find any such movie in the links provided.
Author Response
We thank all of the reviewers for their feedback on our manuscript. We have now addressed all of their concerns. A point-by-point response for each of their concerns (in italics) is included below.
Response to Referee #3
I would recommend rewording the abstract since it is difficult to follow for someone who is not in the field. They need to provide more background information in the abstract.
We have reworded the abstract.
The overlay of the current RssB structures with previous RssB structures (Figures S1 and S4) should be part of the main figures. This is important for the reader to follow the reasoning of the manuscript.
We have moved Figure S1 to the main text new Figure 3 c and d (as suggested), however we don’t believe that it is useful to move Figure S4 into the main text.
Figure 1c, lanes 5 and 6 – Is ClpX being cleaved at 30 and 60 min? If so, this observation should be discussed. Error bars are not clear in Fig. 1d – make them black, for example.
This “clipping” of the ClpX has been observed previously by Houry and colleagues (Thibault et al., 2006), we have now added a brief comment about this clipping and cited the aforementioned publication. To better visualise the error bars (in Fig 1d and 1a) we have decreased the size of the symbol and the thickness of the lines.
In Figure 2a, I am surprised that they see a monomer of ZBD bound to a monomer of RssB. ZBD is a very tight dimer (nM Kd), and I would expect the main species to be a dimer of ZBD and a monomer of RssB. The authors should re‐evaluate their results.
We agree with the reviewer and have re-evaluated our results accordingly and revised the manuscript and figures.
Figure 2d – The authors should explain how the peptides were generated and what is the overlap between the peptides.
This was an oversight and the relevant information has now been added to the Materials and Methods (Section 2.5).
Line 353 – The authors state that no cross‐linking was observed between ZBD and RssBN. This data should be shown.
We have added these data to revised Figure 2 (new Figure 2c)
Lines 357 and 358. Label peptides 20 – 23, 35 – 36, 81 – 82, and 90 – 91 in Figure 2d so readers can assess the significance of these RssB peptides in binding to ZBD.
We have highlighted the “strong” binding peptides and included the sequence of these “strong” binding peptides in revised Figure 2d.
The arrows in Figure 4a showing the directions of rotations do not seem to be correct. Also, label L2 in right most panel.
We have corrected the direction of the arrows. In relation to L2, we have removed our incorrect reference to L2 in the text (line 494).
In the experiments of Fig. 5, does phosphorylated RssBN (or using a phosphor mimic putant) bind σS?
We have not observed a difference in the binding of RssBN in the presence of AcP.
What does the star represent in Fig. 5d?
This indicated a non-specific crosslink between σS and σS, we have included a note in the figure legend
It might be worth to check if the active and inactive conformation od RssB can be distinguished by CD.
This is a useful suggestion, but we are currently unable to perform this analysis at the present time.
Line 571 – It is not clear why the authors mutated K108. A rationale is needed.
A rationale for creating the K108 mutants was briefly described on line 496 and onward.
Why didn’t the authors use their own structure in Figure 6c?
Here we wanted to show the location of the targeted residues in the context of the full-length protein. As we only had structures of each domain (and not the full length protein) we chose to indicate the position of the targeted residues in the context of RssB(full-length) in complex with IraD.
Line 616 – what is the evidence that the R117A mutant reduces the conformational dynamics of RssB?
Currently, we don’t have any direct evidence to suggest that R117A reduces the conformational dynamics of RssB. However, we do have evidence that R117A is not directly involved in SigmaS interaction as this mutant retains the ability to bind to SigS (in the presence of AcP), while in the absence of AcP is unable to interact with SigmaS. Therefore, we speculated that these data suggest that R117 is required to “stabilise” a substrate binding conformation of RssB, which is otherwise stabilised by AcP. We have changed the language in the text to reflect this.
Finally, do the authors have an idea of the binding site of RssBC on ZBD? Identifying such a site would further our understanding of the interactions formed by RssB leading to enhanced σS degradation.
We have preliminary data to suggest that RssB docks into the adaptor binding groove of the ZBD. But this is too preliminary to include in the manuscript.
All minor suggestions have been corrected
Reviewer 4 Report
The work presented by Micevski, et al provides structural and biochemical insights in the molecular mechanism of RssB, the adaptor protein responsible for delivering the master stress response regulator RpoS to ClpXP in E. Coli. The biochemical data they present supports their claims that the zinc-binding domain is necessary for RssB-mediated delivery of RpoS, and that the N-terminus of RpoS is necessary for its delivery to ClpX. There is very nice structural and SAXS data to confirm the assembly of the complexes and the atomic level snapshots of these N- and C-terminal domains of RssB. The structural data is largely consistent with both published and unpublished structures of RssB, with the clear advance of this paper in dissecting the mechanisms of RssB in performing its function. The authors report that the C-terminal domain of RssB seems to bind RpoS and ClpX, that the binding of RssB with ClpX is mediated by the zinc-binding motif, and that phosphorylation of the N-terminal domain affects activity by modulating the RssB conformations. These observations that will add to our understanding of how this critical adaptor works; however, a few additional clarifying experiments, especially about the RssB-ClpX interaction, would strengthen this work.
The authors conclude from Figures 1 and 2 that the C-terminal domain of RssB interacts with the ZBD of ClpX directly. But the only specific data for this direct interaction is a crosslinking experiment (Figure 1) which can have artifacts associated with trapping nonspecific complexes. In particular, if they are saying that the RssB-N does not interact with ZBD, then they should show the data (rather than have it as data not shown). The peptide blot suggests specific regions of RssB are specifically important for binding the ClpX ZBD. The authors should validate this observation with mutations in those regions of RssB that would be predicted to decrease RssB binding to ClpX and its ability to deliver RpoS. If the residues highlighted in Figure 1 are important for ClpX binding, then that should be clear from mutation data.
The authors also focus on the role of the C-terminal domain of RssB in interacting with RpoS. However, the RssB-C binding is much weaker than RssB alone with the biggest effects seen with the crosslinking data. Again, showing the RssB-N data instead of stating data not shown would be important for the crosslinking data. It also seems that simply adding the purified N- or C-terminal domains of RssB and testing competition of RpoS degradation would provide additional needed support to this point.
Finally, the writing overall would be strongly improved if it were broken into smaller paragraphs, for example section 3.6 has several different interesting aspects (showing the role of phosphorylation, mutants, and linker motifs, but these specific details are challenging to parse given the lack of breaks in the text.
Figure Comments:
Figure 1, it would be good to detail the residues associated with the different peptides shown on the blot.
Figure 6. Text says that RE/AA is deficient in RpoS degradation, but this is not shown.
Methods. It would be good to detail what the specific residues contained in the RssB-N and -C constructs (this was in their previous work, but wasn't easily found here).
Discussion. It is unfortunate that none of the putative XBR motifs that they speculated would affect ClpX binding seemed to be important for that - it would be helpful for this to be discussed.
Author Response
Response to Referee #4
In particular, if they are saying that the RssB‐N does not interact with ZBD, then they should show the data (rather than have it as data not shown).
We have included this data. See revised Figure 2c.
The peptide blot suggests specific regions of RssB are specifically important for binding the ClpX ZBD. The authors should validate this observation with mutations in those regions of RssB that would be predicted to decrease RssB binding to ClpX and its ability to deliver RpoS. If the residues highlighted in Figure 1 are important for ClpX binding, then that should be clear from mutation data.
Although mutation of specific regions in RssB could be used to validate the proposed region of interaction, the suggested experiments are not as simple as the reviewer proposes. Unfortunately, our experience with RssB suggests that protein stability/solubility is often affected by mutations within the C-terminal domain, therefore further investigation to validate this proposed binding mechanism site represents a new study, which is beyond the scope of our current manuscript.
The authors also focus on the role of the C‐terminal domain of RssB in interacting with RpoS. However, the RssB‐C binding is much weaker than RssB alone with the biggest effects seen with the crosslinking data.
This is correct, the C-domain binds SigS with reduced affinity (in comparison to full-length RssB). In the manuscript, we have speculated that this may be due to a “secondary” role of the N-domain in regulating the “conformation” of the C-domain (for interaction with SigS).
Again, showing the RssB‐N data instead of stating data not shown would be important for the crosslinking data. It also seems that simply adding the purified N‐ or C‐terminal domains of RssB and testing competition of RpoS degradation would provide additional needed support to this point.
In this case, the reviewer is mistaken, we have not stated “data not shown” for the crosslinking between RssB-N and SigS. Here, we limited our crosslinking experiment to RssB-C and SigS. This is because we had already shown via two alternative approaches (1. co-immunoprecipitation and 2. pull-down) that the bulk of the interaction between RssB and SigS occurred via the C-domain. Therefore, the crosslinking experiment was simply used to confirm the interaction between RssB-C and SigS (using a third approach).
Finally, the writing overall would be strongly improved if it were broken into smaller paragraphs, for example section 3.6 has several different interesting aspects (showing the role of phosphorylation, mutants, and linker motifs, but these specific details are challenging to parse given the lack of breaks in the text.
We have attempted to address this concern throughout the manuscript, including in section 3.6
Figure 1, it would be good to detail the residues associated with the different peptides shown on the blot.
We assume the reviewer is referring to the peptide library in Figure 2 – we have added this information to Figure 2 and supplementary information (new Table S3).
Figure 6. Text says that RE/AA is deficient in RpoS degradation, but this is not shown.
We have not stated in the text or figure 6 that RE/AA is deficient in RpoS degradation. The only mention of the RE/AA mutant describes changes to substrate binding (as is shown in Figure 6). It is nevertheless, true that the RE/AA mutant (perhaps unsurprisingly) is also defective in delivering SigS to ClpXP for substrate degradation. As this information is not particularly significant – given the RE/AA mutant was simply a stepping-stone to the single point mutant R117A (which is a key finding of this manuscript) we have added the requested information to new Supplementary Figure S3. Importantly, both the binding and turnover defect (of R117A) in the presence and absence of AcP is already present in Figure 6.
Methods. It would be good to detail what the specific residues contained in the RssB‐N and ‐C constructs (this was in their previous work, but wasn't easily found here).
Despite the fact that this information is available in a previous publication, we have also included it here (in the Methods section) for easy access, as requested by the reviewer.
Discussion. It is unfortunate that none of the putative XBR motifs that they speculated would affect ClpX binding seemed to be important for that ‐ it would be helpful for this to be discussed.
We have added some discussion regarding the XBR motif – the main point is that in the case of RssB – the putative XBR motif is integral to the structure of RssB (in contrast to the exposed XBR of SspB) and hence is seemingly unavailable for docking to ClpX.
Round 2
Reviewer 4 Report
The authors have addressed all my concerns.